# Single amino-acid mutation in a *Drosoph ila melanogaster* ribosomal protein: An insight in uL11 transcriptional activity

Héloïse Grunchec[1‡], Jérôme Deraze[1‡], Delphine Dardalhon-Cuménal[1], Valérie Ribeiro[1], Anne Coléno-Costes[1], Karine Dias[2], Sébastien Bloyer[3], Emmanuèle Mouchel-Vielh[1], Frédérique Peronnet[1‡*], Hélène Thomassin[1‡*]

1 Laboratoire de Biologie du développement (LBD), Institut de Biologie Paris Seine (IBPS), Centre National de la Recherche Scientifique (CNRS), Sorbonne Université, Paris, France, 2 Genomics Core Facility, Institut de Biologie de l'ENS (IBENS), Département de biologie, École normale supérieure, CNRS, Inserm, Université PSL, Paris, France, 3 Institute for Integrative Biology of the Cell (I2BC), Université Paris-Saclay, CEA, CNRS, Gif-sur-Yvette, France

‡ HG and JD are considered as co-first authors to this work. FP and HT are considered as co-last and co-corresponding authors to this work.
* frederique.peronnet@sorbonne-universite.fr (FP); helene.thomassin-bourrel@sorbonne-universite.fr (HT)

**Data Availability Statement:** The RNA-Seq gene expression data and raw fastq files are available at the GEO repository under accession number

## Abstract

The ribosomal protein uL11 is located at the basis of the ribosome P-stalk and plays a paramount role in translational efficiency. In addition, no mutant for *uL11* is available suggesting that this gene is haplo-insufficient as many other *Ribosomal Protein Genes* (*RPGs*). We have previously shown that overexpression of *Drosophila melanogaster* uL11 enhances the transcription of many *RPGs* and *Ribosomal Biogenesis genes* (*RiBis*) suggesting that uL11 might globally regulate the level of translation through its transcriptional activity. Moreover, uL11 trimethylated on lysine 3 (uL11K3me3) interacts with the chromodomain of the Enhancer of Polycomb and Trithorax Corto, and both proteins co-localize with RNA Polymerase II at many sites on polytene chromosomes. These data have led to the hypothesis that the N-terminal end of uL11, and more particularly the trimethylation of lysine 3, supports the extra-ribosomal activity of uL11 in transcription. To address this question, we mutated the lysine 3 codon using a CRISPR/Cas9 strategy and obtained several lysine 3 mutants. We describe here the first mutants of *D. melanogaster uL11*. Unexpectedly, the *uL11^{K3A}* mutant, in which the lysine 3 codon is replaced by an alanine, displays a genuine *Minute* phenotype known to be characteristic of *RPG* deletions (longer development, low fertility, high lethality, thin and short bristles) whereas the *uL11^{K3Y}* mutant, in which the lysine 3 codon is replaced by a tyrosine, is unaffected. In agreement, the rate of translation decreases in *uL11^{K3A}* but not in *uL11^{K3Y}*. Co-immunoprecipitation experiments show that the interaction between uL11 and the Corto chromodomain is impaired by both mutations. However, Histone Association Assays indicate that the mutant proteins still bind chromatin. RNA-seq analyses from wing imaginal discs show that Corto represses *RPG* expression whereas very few genes are deregulated in *uL11* mutants. We propose that Corto, by repressing *RPG*

GSE181926 (https://www.ncbi.nlm.nih.gov/geo/query/acc.cgi?acc=GSE181926).

**Funding:** FP: Centre National de la Recherche Scientifique (CNRS), Sorbonne University, Fondation ARC grant (PJA20171206407) HG: Doctoral fellowship from the MESRI and Fondation ARC (ARCDOC42020020001381) JD: Doctoral fellowship from the MESRI and Fondation pour la Recherche médicale (FDT20160435164) The funders had no role in study design, data collection and analysis, decision to publish, or preparation of the manuscript.

**Competing interests:** The authors have declared that no competing interests exist.

expression, ensures that all ribosomal proteins are present at the correct stoichiometry, and that uL11 fine-tunes its transcriptional regulation of *RPGs*.

## Introduction

*Drosophila Minute* mutants have been studied for almost a hundred years. They were first described for displaying thin and short bristles, *i.e. Minute* bristles, together with prolonged development [1]. All *Minute* mutations are dominant and lethal when homozygous. The vast majority of them strongly impact viability and fertility, to the point that some could only be identified through transient aneuploidy experiments. *Minute* loci have been characterized over time and all but a few of *Minute* genes have been identified as *Ribosomal Protein Genes* (*RPGs*) [2, 3].

A substantial part of RPs' contribution to cell metabolism has been attributed to their ability to alter ribosome behaviour with consequences on protein synthesis. However, some free RPs are also known for a long time to carry regulatory activities, consequently termed "extra-ribosomal functions" (for a review see [4]). This is notably the case for RPL12, aka uL11 following the new nomenclature proposed to avoid confusion between species [5]. Indeed uL11 was shown in *C. elegans* and mammals to bind its own messenger RNA and inhibit its splicing, leading to aberrant transcripts targeted for degradation through nonsense-mediated mRNA decay [6, 7]. Furthermore, in *S. cerevisiae*, uL11 was shown to be required for the transcription of a subset of PHO pathway genes that are inducible under low phosphate conditions [8].

*Drosophila melanogaster* uL11 is encoded by a unique gene (*uL11/RPL12/ CG3195/ FBgn0034968*) located on the right arm of chromosome 2 at cytogenetic position 60B7. Three transcripts corresponding to *uL11* encode the same 165 amino acid protein. *uL11* expression is ubiquitous and described as "very high" to "extremely high" in all tissues, developmental stages, and cell lines [9]. Two deletions encompassing the whole *uL11* area have been described [*i.e. Df(2R)bw^{VDe2}LPx^{KR}* and *Df(2R)Exel6081*]. However, it was recently proposed that the cytological borders of the first one did not cover the *uL11* region (FBrf0230794) and the second one was lost (FBrf0206661). Thus, there is no evidence that aneuploidy at this locus can be viable, and the locus is indeed described as haplolethal [4, 10, 11]. Another *RPG* (*eL39/RpL39*), and several essential genes (*eEF5, yki*) are found in the vicinity of *uL11* [11, 12]. For this reason, the genes responsible for this haplolethality remain uncertain. However, it is likely that *uL11* contributes to this phenotype as no classical allele has been described yet. Moreover, we observed that ubiquitous RNAi-mediated *uL11* inactivation is lethal during the first larval instar.

The uL11 protein forms, together with the ribosomal protein uL10, the basis of the P-stalk, a lateral protuberance of the 60S subunit which is a critical element of the ribosomal GTPase-associated center known to interact with factors involved in translational elongation and termination [13]. The uL11 protein consists of two domains connected with a hinge, a C-terminal globular domain anchored to a conserved region of the 28S rRNA and a flexible N-terminal domain. The N-terminal domain interacts with several translation factors, notably eEF2 [13, 14]. As a consequence, uL11 has been shown to play an important role at many steps of the translational cycle. In yeast, a deficiency for *uL11* prevents the release of the ribosome associated protein Tif6, which is the last maturation step before the 60S subunit becomes functional and uL11 deficient ribosomes display a decrease in translational fidelity [15]. In cultured human cystic fibrosis bronchial epithelial cells, depletion of uL11 reduces the rate of translational initiation and elongation [16].

We have recently established that *Drosophila melanogaster* uL11 interacts with the chromodomain of the epigenetic co-factor Corto [17]. Chromodomains recognize trimethylated lysines on histones, and the interaction between uL11 and Corto was shown to require the trimethylation of uL11 on lysine 3 (uL11K3me3). Trimethylation of uL11 lysine 3 is very well conserved, existing also in *S. cerevisiae*, *S. pombe* and *A. thaliana* [18–20]. In *D. melanogaster*, uL11 and Corto bind chromatin, colocalize at many sites on polytene chromosomes and are recruited on the *hsp70* gene upon transcriptional activation [17, 21]. Overexpression of uL11 as well as that of the Corto chromodomain induces the transcription of many *RPGs* and *Ribosomal Biogenesis* genes (*RiBis*) suggesting that uL11 could globally regulate the level of translation through its transcriptional activity. Thus, our previous study led us to hypothesize that uL11K3me3 was involved in transcriptional regulation.

To test this hypothesis, we designed a CRISPR/Cas9 strategy for mutating the uL11 lysine 3 codon. In this study we have replaced the lysine 3 codon of uL11 with codons for amino acids that are not subject to methylation; alanine (uL11K3A) and tyrosine (uL11K3Y). The *uL11*$^{K3A}$ mutant, but surprisingly not the *uL11*$^{K3Y}$ mutant, displays *Minute* phenotypes *i.e.* longer development, low fertility, high lethality and thin bristles. As expected, uL11K3A and uL11K3Y proteins do not interact with the Corto chromodomain anymore. However, they still bind chromatin. RNA-seq analyses from wing imaginal discs show that Corto represses *RPG* expression whereas very few genes are deregulated in *uL11* mutants.

## Materials and methods

### *Drosophila* genetics

*Drosophila melanogaster* stocks and crosses were kept on standard yeast corn-meal medium (7.5% yeast, 9% cornmeal, 1% agar, 0.27% Moldex) at 25˚C. For all experiments, crosses were set up with similar densities to prevent confounding effects of overcrowding. $w^{1118}$ was used as the control strain. The *corto*$^{420}$ and *corto*$^{L1}$ loss-of-function alleles were described in [17].

### Strategy of CRISPR/Cas9 mutagenesis

The *pU6-chiRNA:sgRNA* plasmid was obtained by incorporating the *sgRNA* sequence (obtained by annealing *phos-gRNA-F* and *phos-gRNA-R*, S1 Table) into *pU6-BbsI-chiRNA* (Addgene plasmid # 45946) [22, 23] following the protocol provided on https://flycrispr.org/protocols/. Sequence was confirmed using the T3 universal primer.

A 123 nucleotide-long single-stranded oligodeoxynucleotide (*ssODN*) carrying the lysine (AAA) to alanine (GCC) substitution flanked by two 60 nucleotide-long homology arms was used as a template for Homology-Directed Repair (HDR) (synthetized by Integrated DNA Technologies Inc) (S1 Table). The *uL11* region of the recipient line *vasa-Cas9* (BL-51324) was sequenced in order to respect possible polymorphisms. To prevent base pairing with the *sgRNA*, the *ssODN* was designed to be homologous to the PAM carrying strand.

### Fly transgenesis

Two hundred *vasa-Cas9* embryos were injected with a mixture containing 100 ng/μL *pU6-chiRNA:sgRNA* and 100 ng/μL *ssODN* (BestGene Inc.). Transformant G0 flies (48 females and 44 males) were individually crossed to $w^{1118}$; *In(2LR)Gla, wgGla1/SM5* flies (*Gla/SM5*). Among them, only 18 males and 11 females were fertile. Curly wing G1 siblings were individually crossed to *Gla/SM5* flies. Once the G2 progeny born, G1 founding flies were harvested, genomic DNA extracted, and LNA allele-specific genotyping performed as described below. 294 G1 individuals were genotyped to detect the presence of *uL11* mutant alleles. Curly wing

G2 offspring of G1 flies carrying a mutant allele of *uL11* were then crossed with each other to establish mutant balanced strains. In order to eliminate potential unspecific mutations, ten balanced mutant females from the offspring were then individually backcrossed with $w^{1118}$ males and genotyped by High Resolution Melting Analysis (HRMA) after laying eggs. *uL11 m*utant females were kept and the whole procedure was repeated seven times.

### Genomic DNA extraction

Genomic DNA was extracted by crushing single flies in 100 μL SB buffer (10 mM Tris pH 8.0, 1 mM EDTA, 25 mM NaCl, 200 μg/μL Proteinase K), followed by 30 min incubation at 37˚C. DNA was further purified by standard phenol-chloroform extraction followed by ethanol precipitation.

### Genotyping by locked nucleic acid allele-specific qPCR

Forward allele-specific primers with 3' end matching either wild-type (*LNAWT*) or mutated 3rd codon (*LNAK3A*) of *uL11* (AAA or GCC, respectively) and a Locked Nucleic Acid (LNA) nucleotide [24] at the second position of the mismatch codon were used in combination with a reverse primer (*CRISPR1_R*) to amplify a 219 nucleotide fragment (S1 Table). 25 μL reactions were set to contain 5 to 15 ng of genomic DNA, 0.5 μM forward and reverse primers, 0.4 nM dNTP, 0.75 μL SYBR green (Diagenode), and 2.5 units of DreamTaq polymerase (Thermo Fisher Scientific) in TMAC buffer (67 mM Tris pH 8.8, 6.7 mM MgCl2, 16.6 mM ammonium sulfate, 0.5 mM tetramethylammonium chloride, 0.17 mg/mL BSA) [25]. 0.5 ng of plasmid containing the *uL11* coding region in which the AAA lysine 3 codon was replaced by GCC was used as positive control. qPCR reactions were carried out in a CFX96 system (BioRad) [95˚C 3 min; 40 cycles (95˚C 20 s, 64˚C 20 s, 72˚C 30 s)]. To confirm the presence of the mutated allele, a 1.5 kb region centred on the lysine 3 codon was sequenced.

### High Resolution Melting Analysis (HRMA)

Genomic DNA was analysed by HRMA as described in [26]. Briefly, oligonucleotides *uL11-HRMA-F* and *uL11-HRMA-R* (S1 Table) were used to amplify a 173 bp region centred on the *uL11* lysine 3 codon. PCR reactions were performed with SsoFast™ EvaGreen® Supermix (Bio-Rad) in 20 μL reactions containing 2 to 15 ng genomic DNA and 0.5 μM each oligonucleotide. Cycles were carried out in a CFX96 system (BioRad) [98˚C 3 min; 40 cycles (98˚C 2 s, 57.3˚C 15 s)]. Thermal melting profiles were obtained in the same device by increasing temperature from 75 to 95˚C using a temperature increment of 0.2˚C. They were normalized as described by [27].

### uL11K3me3 antibodies

Polyclonal anti-uL11K3me3 antibodies were generated in rabbit using a peptide corresponding to the first 16 amino acids of uL11 with methylated lysine 3 [PPK(me3)FDPTEVKL-VYLRC] (Eurogentec). The serum was first loaded on a uL11K3me3 peptide affinity column which allowed to retain anti-uL11K3me3 and anti-uL11 antibodies. After elution, they were separated by passage through an unmethylated uL11 peptide affinity column. Specificity of the antibodies was checked by dot blot (S1 Fig).

Proteins were extracted from third instar larvae in RIPA buffer (150 mM sodium chloride, 1% NP40, 0.5% sodium deoxycholate, 0.1% SDS, 50 mM Tris-HCl pH 8,0) supplemented with phosphatase and protease inhibitors (Roche). 30 μg of proteins were separated by SDS-PAGE electrophoresis on 15% acrylamide gels. Western blots were performed according to standard

protocols using either goat anti-uL11 (SantaCruz sc82359, 1/1000), rabbit anti-uL11K3me3 (1/6000), or mouse anti-α-tubulin (DSHB E7c, 1/2500) as primary antibodies. Anti-goat (Jackson ImmunoResearch; 705035147; 1/10000), anti-rabbit (1/20000) or anti-mouse (Sigma NA931; 1/20000) were used as secondary antibodies.

## Analysis of mutant life history traits

uL11 wild-type or mutant chromosomes were balanced with *CyO,Dfd-EYFP* (from strain BL-8578) or SM5. About 100 females and 60 males were placed in laying cages on agarose plates (2% agarose, 5% vinegar, neutral red) supplemented with yeast. To measure embryonic lethality, 100 embryos were collected from each laying cage, transferred on new agarose plates and emerging first instar larvae were counted. To measure larval and pupal lethality, 100 embryos were collected and transferred into yeast cornmeal medium tubes at 25˚C. Pupae and adults were then counted. Three independent experiments were performed and results were pooled. To measure developmental time, first instar larvae were collected and transferred into yeast cornmeal medium tubes at 25˚C (50 to 100 larvae per tube). Vials were checked from 9 days after egg laying until no more adults emerged. Statistical significance was assessed by Chi-2 tests.

## Measure of bristle length

Adult bodies free of wings, legs and heads were aligned on agar cups. Images were captured using a Leica Model MZ FLIII microscope equipped with a Leica Model DC480 camera. Scutellar bristles were measured using the ImageJ segmented line tool. Normality was checked by Shapiro-Wilk tests and homogeneity of variances by F tests. Student's t-tests were then set taking into account homo- or heteroscedasticity of variances.

## Measure of wings

Adult flies were kept in 70% ethanol for 48 h and transferred into PBS glycerol (1:1 v/v). Wings were dissected and mounted on glass slides, dorsal side up, in Hoyer's medium. Slides were scanned with a Hamamatsu Nanozoomer Digital Slide scanner, running the Nanozoomer software with a 20x objective and an 8 bit camera. Wing pictures were separately exported into TIF format using NDP.view and the 5 x lens. Measurements of wing length were performed as described in [28].

## Plasmids

*uL11* was amplified from *w^1118^* embryonic cDNAs and subcloned into *pENTR/D-TOPO* (Invitrogen) [17]. *pENTR-uL11^K3A^* and *pENTR-uL11^K3Y^* were obtained by site-directed mutagenesis using the oligonucleotides described in [17] and in S1 Table, respectively. The cDNAs were then transferred either into the *pAWM* or the *pAWH* Gateway® *Drosophila* vectors allowing expression of fusion proteins with a C-terminal Myc or HA tag under the control of an actin promoter in S2 cells.

## Cell transfection

S2 cells were cultured at 25˚C in Schneider's *Drosophila* medium supplemented with 10% heat-inactivated fetal bovine serum and 100 units/mL of penicillin and streptomycin (Life technologies). To obtain cells permanently expressing uL11-HA, uL11K3A-HA or uL11K3-Y-HA, a mix containing a 5:1 molar ratio of the *pA-uL11-HA*, *pA-uL11^K3A^-HA* or *pA-uL11^K3Y^-HA* expression vector and the selection plasmid *pCoBlast* (Invitrogen) was prepared.

$10^6$ cells were then transfected with 2 μg of DNA using Effecten® transfection reagent (Qiagen) according to the manufacturer's instructions at a 1:10 DNA/Effecten® ratio. Selection was performed by addition of 10 μg/mL of blasticidin after 48 h. After initial selection, stable cell lines were cultured in presence of 2 μg/mL of blasticidin. For transient expression, $10^6$ cells were transfected with 2 μg of either *pA-uL11-Myc*, *pA-uL11*$^{K3A}$*-Myc* or *pA-uL11*$^{K3Y}$*-Myc*, and 2 μg of *pA-FLAG-CortoCD* [17] using Effecten® at a 1:10 DNA/Effecten® ratio.

## Polysome fractionation

Cells were harvested at 50% confluence and washed in Schneider medium at room temperature to remove the fetal bovine serum. They were then resuspended in ice-cold lysis buffer (20 mM Hepes pH 7.5, 250 mM KCl, 10 mM MgCl2, 5 mM DTT, 1 mM EDTA, 0.5% NP-40) supplemented with EDTA-free protease inhibitor cocktail (Roche Diagnostics) and 40 U/mL Ribolock RNAse Inhibitor (ThermoFisher). For EDTA treatment, the same buffer adjusted to 25 mM EDTA and without MgCl2 was used. After centrifugation at 500 g for 5 min to pellet nuclei, supernatants were layered onto 10 to 50% sucrose gradients in polyribosome buffer (20 mM Hepes pH 7.5, 250 mM KCl, 20 mM MgCl2, 2 mM DTT), supplemented with EDTA-free protease inhibitor cocktail and 40 U/mL Ribolock RNAse Inhibitor. Gradients were centrifuged at 39,000 rpm for 165 min at 4˚C in a Beckman SW41-Ti rotor. Optical density at 254 nm was monitored using a density gradient fractionator (Teledyne Isco, Lincoln, NE).

Western blots were performed according to standard protocols using mouse anti-HA (Sigma F2411; 1/1000) as primary antibodies and anti-mouse (Sigma NA931; 1/20000) as secondary antibodies. They were revealed using the Supersignal™ West Pico PLUS Chemiluminescent Substrate (Thermo Scientific) as described by the supplier.

## Puromycin assays

Puromycin assays were adapted from [29] with the following modifications: 20 third instar larvae were turned inside-out and incubated for 1 h at 25˚C under gentle rotation in Schneider's medium supplemented or not with 10 mg/mL cycloheximide (Sigma). Puromycin (antpr1, InvivoGen) was then added at a final concentration of 0.28 mg/mL and incubation was continued for 2 h. Total proteins were extracted in a buffer containing 30 mM Hepes pH 7.4, 0.1% NP40, 150 mM NaCl, 2 mM Mg(OAc)$_2$ supplemented with phosphatase and protease inhibitors (Roche) (adapted from [30]). 60 μg of protein extracts were deposited on a 12% acrylamide gel.

Western blot were performed according to standard protocols using mouse anti-puromycin (Kerafast, 3RH11; 1/500) or mouse anti-H3 (Diagenode; C15200011; 1/1000) as primary antibodies, and anti-mouse (Sigma; NA931; 1/20000) as secondary antibodies and revealed using the Supersignal™ West Pico PLUS Chemiluminescent Substrate (Thermo Scientific). Puromycin and H3 signals were measured using ImageJ. The puromycin signal (signal in the samples treated with CHX and puromycin minus signal in the untreated sample) was normalized towards the H3 signal. Statistical significance was assessed by Student's t-tests.

## Co-immunoprecipitation

S2 cells transiently transfected with either *pA-uL11-Myc*, *pA-uL11*$^{K3A}$*-Myc* or *pA-uL11*$^{K3Y}$*-Myc* and *pA-FLAG-CortoCD* [17] were harvested after 48 h and washed in Schneider medium at room temperature. Co-immunoprecipitations were performed as described in [31] without fixation. 30 μl of Protein G coated Bio-Adembeads (Ademtech) were incubated with either 1 μg of mouse monoclonal anti-FLAG antibody (F3165, Sigma), or goat anti-HA antibody as mock (sc-805, Santa Cruz Biotechnology).

Western blot were performed according to standard protocols using mouse anti-Myc (Abcam ab9132; 1/10000) or anti-FLAG (Sigma F3165; 1/5000) as primary antibodies and anti-mouse (Sigma NA931; 1/20000) as secondary antibodies and were revealed using the Supersignal™ West Pico PLUS Chemiluminescent Substrate (Thermo Scientific).

### Histone association assays

S2 cells permanently expressing uL11-HA, uL11K3A-HA or uL11K3Y-HA were cross-linked with formaldehyde and chromatin was prepared as described [32]. One μL of chromatin was kept for the input and immunoprecipitation was performed on the remaining chromatin using anti-H3 antibodies (mouse monoclonal, C15200011, Diagenode; 3 μg/IP). Chromatin-associated proteins were detected by Western blot: H3 was detected using anti-H3 antibodies (1/5000), uL11-HA proteins using anti-HA antibodies (Sigma F2411; 1/1000), and histone H2B using anti-histone H2B antibodies (Abcam ab1790; 1/2000); anti-mouse (Sigma NA931; 1/20000) or anti-rabbit (A0545, Sigma; 1/20000) were used as secondary antibodies. Western blots were revealed with a ChemiDoc MP Imaging System (Bio-Rad).

### RNA-seq, bioinformatic analyses and RT-qPCR

Wing imaginal discs of third instar female larvae (one disc per larva) were dissected by batches of 50 in ice-cold PBS and frozen in liquid nitrogen. 150 discs (three batches) were pooled. Total RNAs were extracted using RNeasy kit (Qiagen).

Preparation of library and RNA-seq from *corto*$^{420}$/*corto*$^{L1}$ wing imaginal discs were performed as described in [17]. For *uL11* wing imaginal discs, library preparation was performed using the TruSeq® Stranded mRNA Library Prep kit (Illumina). Library preparation and Illumina sequencing were performed at the École Normale Supérieure genomics core facility (Paris, France) on a NextSeq 500 (Illumina). Three replicates were sequenced for each genotype. 75 bp single reads were trimmed using FastQC (Galaxy version 0.72). Reads were then aligned against the *D. melanogaster* genome (dm6 genome assembly, release 6.30) using STAR (Galaxy Version 2.6.0b-2). Reads were counted using FeatureCounts (Galaxy Version 1.6.0.6). Differential analysis was performed using DESeq2 version 1.32.0. Gene ontology was analysed with DAVID (https://david.abcc.ncifcrf.gov/home.jsp). The RNA-Seq gene expression data and raw fastq files are available at the GEO repository (https://www.ncbi.nlm.nih.gov/geo/info/seq.html) under accession number GSE181926.

RT-qPCR were performed on wing imaginal disc cDNAs as described in [28]. Expression levels were quantified with the Pfaffl method [33] and normalized to the geometric mean of two reference genes, *GAPDH* and *Spt6*, the expression of which did not vary in the mutants. Sequences of primer pairs are listed in S1 Table.

## Results

### CRISPR/Cas9 editing of the uL11 lysine 3 codon

*uL11* is located within a cluster of highly transcribed genes, many of which are also essential (including *eEF5*, *RpL39/eL39*, *yki*...) [11, 12]. This cluster is indeed part of the 1.6% haplolethal regions of the euchromatic *Drosophila* genome [3]. Furthermore, *uL11* is bordered by two small intergenic sequences (465 and 620 bp, respectively) that might contain regulatory elements (Fig 1). Thus, the insertion of a selection cassette within this locus could disrupt gene expression and impede viability. We therefore chose to edit the *uL11* lysine 3 to alanine (K3A) by a single step CRISPR/Cas9 mediated HDR using a single-stranded oligodeoxynucleotide donor template (*ssODN*) [23, 34].

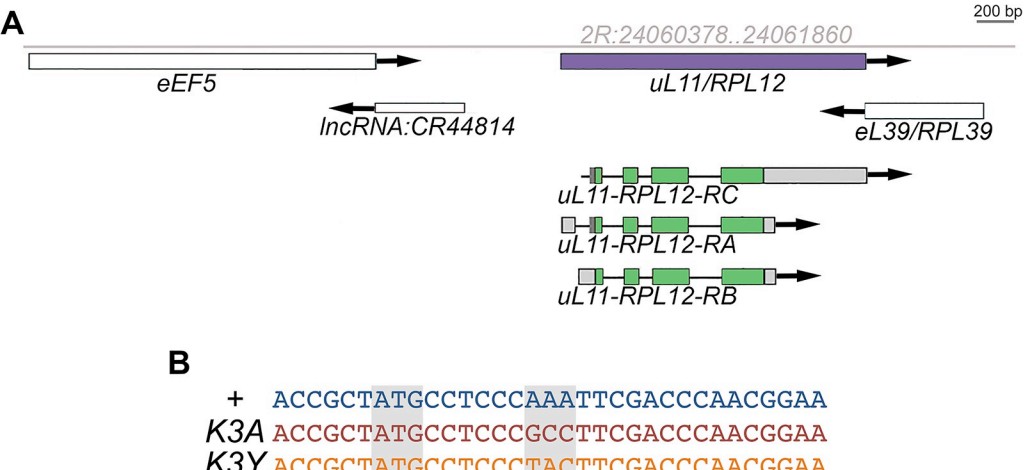

**Fig 1. Genomic organization of the *uL11* locus.** A–Genomic locus of the 2R chromosome containing the *uL11* gene (coordinates in grey). B–Sequence of the *uL11*$^{K3A}$ and *uL11*$^{K3Y}$ mutants.

To recover the successful HDR events, we set up a molecular screening protocol based on an allele specific amplification strategy [34]. Discriminating power was increased by the substitution of the penultimate nucleotide of the screening primers with a locked nucleic acid (LNA) (S2 Fig). The presence of a single LNA sufficiently improved specificity to allow us analyzing pools of flies for the presence of a single allele copy (S2 Fig). 294 G1 individuals were tested for the presence of a mutated allele carrying the K3A substitution. Mixtures of genomic DNA from pools of 4 to 5 individuals were prepared and the *uL11* locus was amplified with either the lysine codon (*LNA-WT*) or the alanine codon (*LNA-K3A*) matching primer. While most genomic DNA mixtures displayed amplification kinetics similar to the one of the negative control, six of them exhibited faster amplification (ΔCt between 2 and 7). We thus repeated the experiment on individual genomic DNAs from 6 positive pools. Ten genomic DNA originating from three independent G0 founding flies, two males and one female, exhibited quicker amplification with the *LNA-K3A* primer than with the control primer (ΔCt>5) (S2 Fig). Sequencing the *uL11* locus confirmed that these flies were heterozygous for the recombinant allele *uL11*$^{K3A}$. Three *uL11*$^{K3A}$ mutants, *uL11*$^{K3A-12}$, *uL11*$^{K3A-43}$ and *uL11*$^{K3A-6}$, coming from three different founders, were obtained (S3 Fig). Unless specified, analyses were performed with *uL11*$^{K3A-43}$.

To detect other mutations potentially resulting from non-homologous end joining (NHEJ) events, we also performed HRMA of a qPCR amplicon centred on the *uL11* lysine 3 codon. Denaturation kinetics of these PCR products were analyzed individually for the 294 G1 flies. Among them, 36 denaturation profiles differed from the wild-type control. Sequencing of the amplicons confirmed the presence of a mutation at the *uL11* locus in each of these 36 samples. Consistently, the 10 *uL11*$^{K3A}$ mutants identified with the allele-specific amplification strategy were also recovered by HRMA. Seven additional alleles were thus identified, that all carry a mutation impairing the lysine 3 codon: a single (K3Y) or double (P2LK3E, P2QK3R) amino acid substitutions, a single (ΔK3) or double (ΔK3F4) amino acid deletion, and an insertion of 2 or a deletion of 4 nucleotides (F+2 and F-4, respectively) (S3 Fig). Preliminary observations revealed that the mutants could be dispatched into two groups depending on the severity of their phenotypes. The first group contains the *K3A*, *ΔK3*, *ΔK3F4*, *F+2* and *F-4* alleles, and the second group the *K3Y*, *P2QK3R* and *P2LK3E* alleles. We choose to focus on two representative alleles, *K3A* and *K3Y* (Fig 1 and S2 and S3 Figs). They were introduced into the same

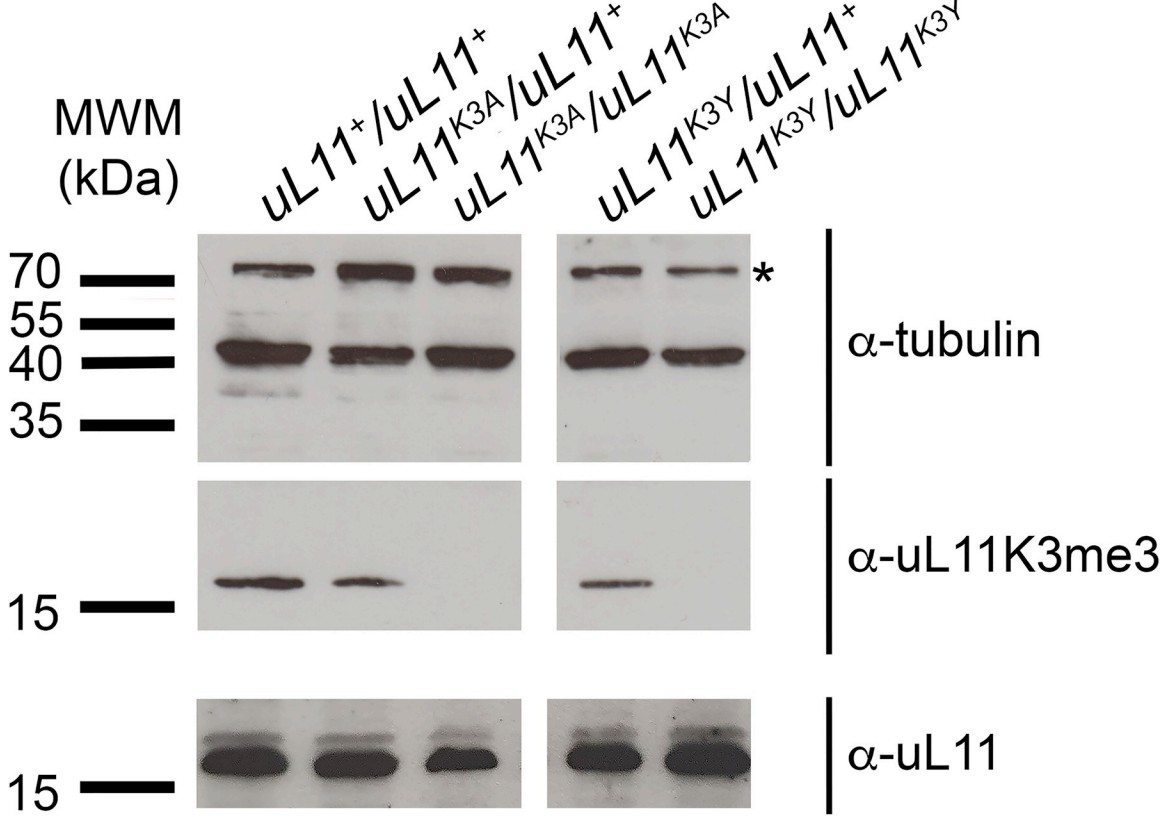

**Fig 2. uL11K3me3 is undetectable in homozygous *uL11* mutants.** Tubulin was used as a loading control. Whereas uL11 was present in hetero- and homozygous mutants, as revealed by the pan-uL11 antibody, uL11K3me3 was undetectable in the two homozygous mutants. *: unspecific signal.

controlled genetic background ($w^{1118}$) and expression of the uL11 proteins was confirmed by western blot (Fig 2). As expected, uL11K3me3 was not detected in the homozygous *K3A* and *K3Y* mutants.

## Lethality and developmental delay of *uL11* mutants

We first examined the lethality of *uL11* homozygous mutants. Whereas we obtained many homozygous *uL11*$^{K3Y}$ adults, very few *uL11*$^{K3A}$ homozygotes emerged and almost all of them were males. To follow the lethality of *uL11*$^{K3A}$ mutants during development, we compared the number of embryos, larvae and pupae to the one of the $w^{1118}$ control. During embryogenesis, they did not display more lethality than the $w^{1118}$ control with the exception of *uL11*$^{K3A}$/*uL11*$^+$ whose lethality is slightly higher (Chi$^2$ test, $p < 0.05$) (Fig 3). By contrast, during larval life, lethality was very high for *uL11*$^{K3A}$/*uL11*$^+$, and *uL11*$^{K3A}$/*uL11*$^{K3A}$ (Chi$^2$ test, $p < 0.001$), but did not increase neither for *uL11*$^{K3Y}$/*uL11*$^+$ nor for *uL11*$^{K3Y}$/*uL11*$^{K3Y}$ larvae. We did not observe any lethality during the pupal life for all genotypes. Similarly, developmental time from egg deposition to adult emergence was considerably extended for *uL11*$^{K3A}$/*uL11*$^+$ (up to 48 h) and *uL11*$^{K3A}$/*uL11*$^{K3A}$ (up to 96 h) and for a second *uL11*$^{K3A}$ mutant, *uL11*$^{K3A-12}$, either heterozygotes or homozygotes (S4 Fig). However, the developmental time of *uL11*$^{K3Y}$/*uL11*$^{K3Y}$ flies was unaffected (Fig 3).

To summarize, the *K3Y* mutation had no effect on these life history traits whereas lethality and developmental time were increased both in heterozygous and homozygous *K3A* mutants, which characterized this allele as dominant.

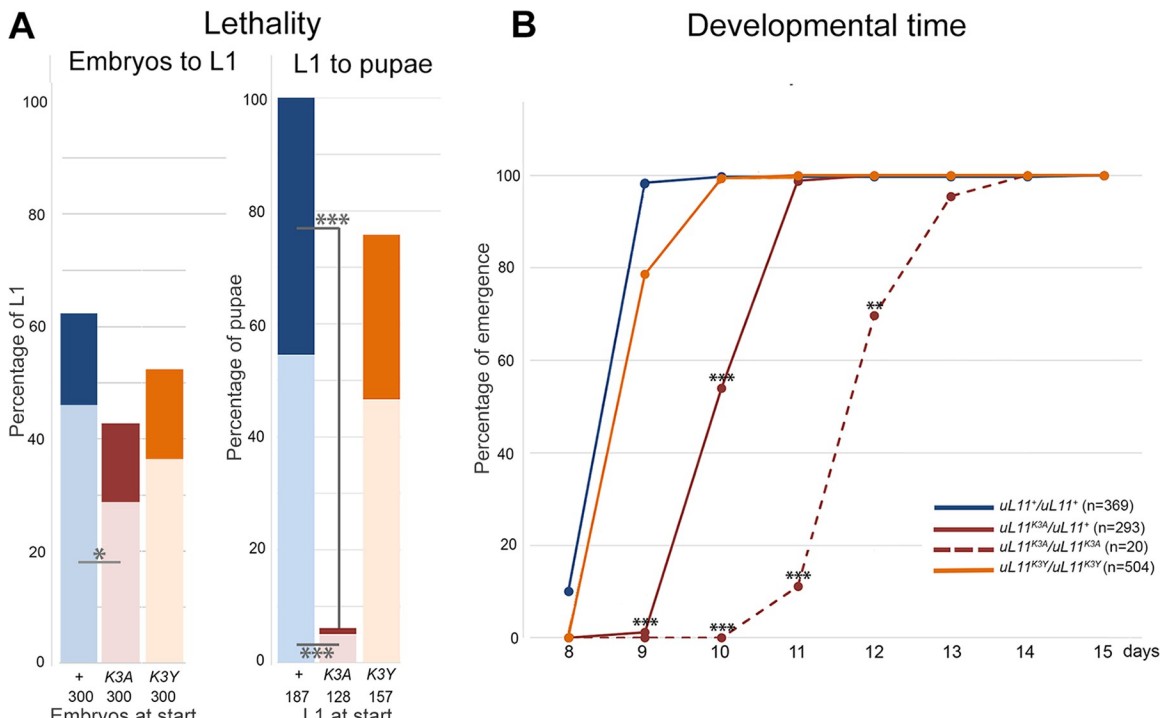

**Fig 3. Life history traits of *uL11* mutants.** A–Lethality of *uL11* mutants. *uL11$^X$/CyO, Dfd-EYFP* flies were crossed between them and 300 embryos per genotype were transferred on new medium (*uL11$^X$*: *uL11$^+$*, *uL11$^{K3A}$* or *uL11$^{K3Y}$*, as indicated). Left: percentage of *uL11$^X$/uL11$^X$* and *uL11$^X$/CyO,Dfd-EYFP* first instar larvae emerged from 300 embryos (*CyO,Dfd-EYFP* homozygous embryos did not emerge); Right: percentage of *uL11$^X$/uL11$^X$* and *uL11$^X$/CyO,Dfd-EYFP* pupae coming from the emerged first instar larvae (numbers of starting embryos and larvae are indicated). *uL11$^X$/ uL11$^X$* individuals: dark colour; *uL11$^X$/CyO,Dfd-EYFP* individuals: light colour; blue: *uL11$^+$*, burgundy: *uL11$^{K3A}$*, orange: *uL11$^{K3Y}$*. B–Developmental time of uL11 mutants. The percentage of flies emerged from day 8 to 15 is shown. The total number of emerged flies is indicated. Solid lines: heterozygous flies; dotted line: homozygous flies; blue: *uL11$^+$*, burgundy: *uL11$^{K3A}$*, orange: *uL11$^{K3Y}$*. Chi$^2$ test: *** p-value < 0.001; ** p-value < 0.01; * p-value < 0.05; only significant comparisons are shown.

## Bristle and wing size of the *uL11* mutants

*Minute* mutants have thinner and shorter bristles as compared to wild-type flies. As it also seemed to be the case for *uL11$^{K3A}$* mutants either heterozygotes or homozygotes (Fig 4 and S4 Fig), we measured the anterior and posterior scutellar bristles of *uL11$^{K3A}$* and *uL11$^{K3Y}$* mutants. In males, scutellar bristles were indeed significantly shorter in heterozygous and homozygous *uL11$^{K3A}$* mutants as compared to control flies but unaffected in homozygous *uL11$^{K3Y}$* mutants (Fig 4). In females, we also observed shorter bristles in heterozygous and homozygous *uL11$^{K3A}$* mutants while bristles of *uL11$^{K3Y}$* homozygous mutants were only slightly affected (S5 Fig). Furthermore, heterozygous and homozygous *uL11$^{K3A}$* males have shorter wings while those of *uL11$^{K3Y}$* homozygous males were unaffected (Fig 4). Similarly, heterozygous *uL11$^{K3A}$* females exhibited shorter wings whereas wings of homozygous *uL11$^{K3Y}$* females were unaffected (S5 Fig).

These results confirmed that *uL11$^{K3A}$* exhibits characteristics of a dominant allele and showed that the severity of the phenotypes depended on the mutation. The *K3A* mutation was highly detrimental while the *K3Y* mutation had almost no impact on the size of scutellar bristles and wings. *Minute* mutants are known to be poorly fertile and viable, to exhibit developmental delay and have shorter and thinner bristles, all phenotypes that we observed in *uL11$^{K3A}$* mutant flies. Moreover, *Minute* alleles are dominant which is also the case of the *uL11$^{K3A}$* allele. All these data characterized the *uL11$^{K3A}$* mutant as a *Minute* mutant.

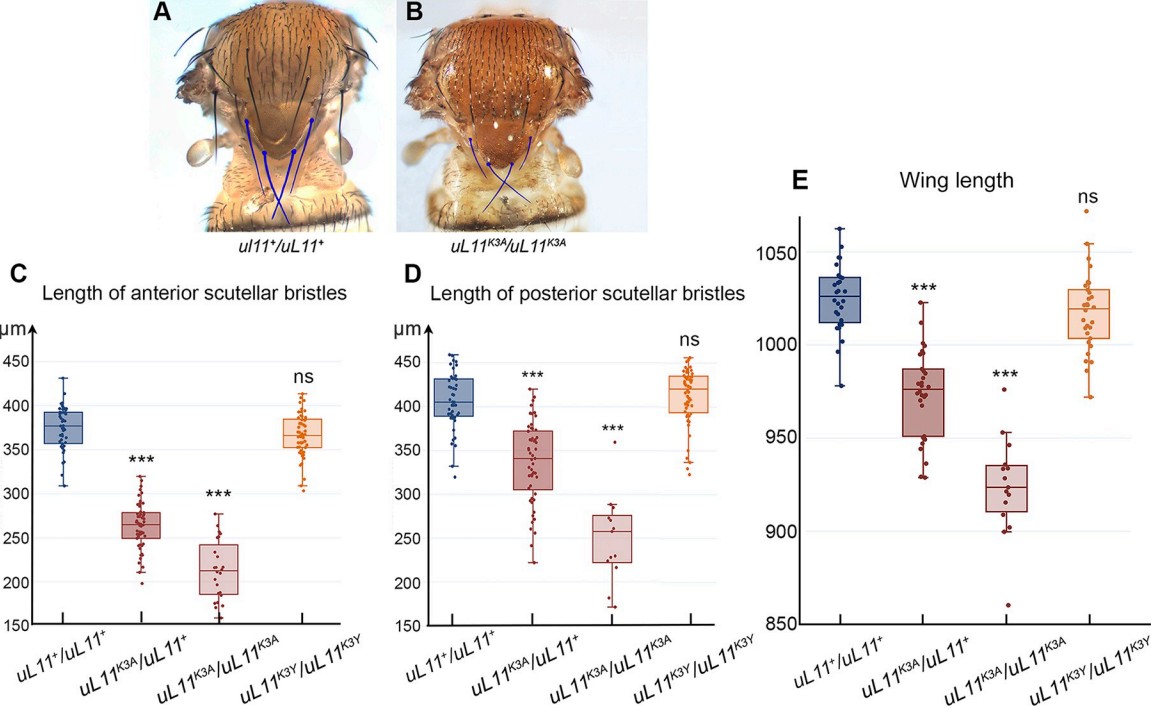

**Fig 4. Analysis of bristles and wings in male *uL11* mutants.** A–Thorax of a wild-type male. Anterior and posterior scutellar bristles are colorized. B–Thorax of a *uL11^{K3A}/uL11^{K3A}* male. Anterior and posterior scutellar bristles are colorized. They look thinner and shorter that those of the wild-type male shown in A. C–Length of anterior scutellar bristles of wild-type males (blue; n = 45), *uL11^{K3A}/uL11^{+}* and *uL11^{K3A}/uL11^{K3A}* (dark and light burgundy, n = 55 and n = 24, respectively) and *uL11^{K3Y}/uL11^{K3Y}* (orange, n = 51). D–Length of posterior scutellar bristles of wild-type males (blue; n = 44), *uL11^{K3A}/uL11^{+}* and *uL11^{K3A}/uL11^{K3A}* (dark and light burgundy, n = 53 and n = 13, respectively) and *uL11^{K3Y}/uL11^{K3Y}* (orange, n = 50). E–Wing length of uL11 wild-type males (blue; n = 28), *uL11^{K3A}/uL11^{+}* and *uL11^{K3A}/uL11^{K3A}* (dark and light burgundy, n = 30 and n = 15, respectively) and *uL11^{K3Y}/uL11^{K3Y}* (orange, n = 30). t-tests: *** p-value < 0.001; ns: non significant.

## Impact of the *uL11* mutations on translation

The strategic location of uL11 protein at the basis of the P-stalk in the GTPase-associated center of the ribosome suggests that its mutation might have a detrimental impact on translation. To test this hypothesis, we assessed the level of global translation in the uL11 mutants. In order to label neo-synthesized proteins, third instar wild-type or mutant larvae were incubated with puromycin. Puromycin intake was normalized to histone H3 levels. A significant decrease in global translation level was observed in *uL11^{K3A}* homozygous larvae as compared to wild-type larvae, whereas it was unmodified in *uL11^{K3Y}* homozygous, in agreement with the absence of *Minute* phenotypes (Fig 5). *uL11^{K3A}* heterozygous larvae, which however present *Minute* phenotypes, did not exhibit any decrease in the global translation level. On the one hand, the phenotypes of *uL11^{K3A}/ uL11^{+}* flies are not as severe as those of *uL11^{K3A}/ uL11^{K3A}*, and a low decrease in global translation might be undetectable in our assays. On the other hand, the amount of protein synthesis is known to vary depending on the proliferative activity of the tissue or even depending on the cell type [35]. For example, it is visibly impacted by heterozygous *RPG* mutations in clonal analyses of cell competition in *Drosophila* larvae wing imaginal discs [36, 37]. A low decrease of protein synthesis in larval proliferative tissues might be averaged in our global analyses.

We then asked whether the uL11K3A protein would retain the ability to associate with translating ribosomes. We generated stable cell lines expressing either uL11K3A-HA or uL11-HA under the control of the *Actin* promoter. Cytoplasmic extracts were purified from both genotypes and lysates were loaded onto sucrose gradients for fractionation. As a control,

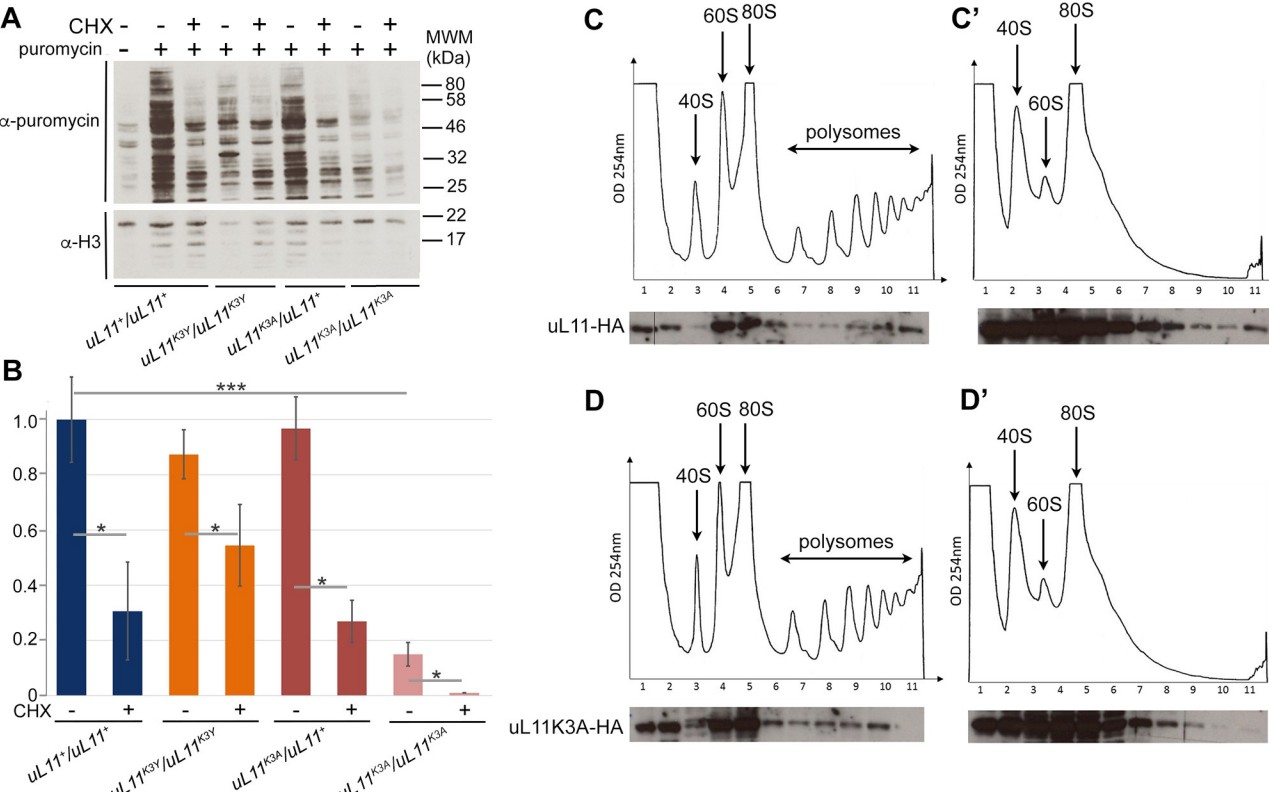

**Fig 5. Translational rate of *uL11* mutants.** A–Western blot showing puromycin incorporation in larvae of the two mutants as compared to wild-type larvae, in absence (-) or presence (+) of cycloheximide (CHX), an inhibitor of translation. Puromycin incorporation was revealed with an anti-puromycin antibody. Histone H3, revealed with an anti-panH3 antibody, was used as loading control. MWM: molecular weight marker. B–Quantification of the puromycin signal in the 4 genotypes without (-) or with (+) CHX treatment. The puromycin signal was normalized to the H3 signal. Student's t-tests were performed to compare puromycin incorporation in mutant and wild-type larvae. * p-value < 0.05; ns: non significant. C, C', D, D'–Polysome fractionation: cytoplasmic lysates (C, D) and EDTA-treated lysates (C', D') from S2 cells expressing uL11-HA (C, C') or uL11K3A-HA (D, D') were fractionated by centrifugation onto a sucrose gradient. Optical density at 254 nm was monitored during fractionation (top panels). The peaks observed in the gradient correspond to the different ribosomal complexes: 40S subunit, 60S subunit, 80S monosome, polysomes. Proteins extracted from fractions were analyzed by Western blotting with anti-HA antibody (lower panels). A vertical line indicates that different wells from the same gel were juxtaposed in the image for clarity. Images are representative for three obtained replicates.

an extract of each genotype was supplemented with 25 mM EDTA, a concentration that disrupts the interaction between ribosomal subunits and mRNA. The resulting fractions were analyzed by western blot to reveal the presence of uL11K3A-HA or uL11-HA (Fig 5). These experiments revealed an enrichment of uL11-HA in the 60S fraction (large ribosomal subunits), in the 80S fraction (ribosomes) and in polysomes, but not in the 40S fraction (small ribosomal subunit), as expected. The same pattern was observed for uL11K3A-HA. Furthermore, EDTA treatment triggered the relocation of uL11-HA and uL11K3A-HA towards lighter fractions, confirming that the sedimentation profiles truly resulted from their association with polysomes (Fig 5). All these data showed that uL11K3A-HA was efficiently incorporated into translating ribosomes. Nevertheless, the decrease in puromycin incorporation observed in $uL11^{K3A}$ mutants suggests that the yield of translation might be altered.

## Interaction of uL11 mutant proteins with Corto and chromatin

Overexpression of uL11 as well as overexpression of the Corto chromodomain enhances the transcription of *RPGs* and *RiBi* genes [17]. As both proteins interact together and bind

chromatin, we hypothesized that their transcriptional activity were linked. We have previously shown that the trimethylation of uL11 on lysine 3 mediates the interaction between uL11 and the chromodomain of the epigenetic cofactor Corto (CortoCD). We first assayed the physical interactions between CortoCD and the mutant uL11 proteins. To do this, we co-transfected *Drosophila* S2 cells with *pA-FLAG-CortoCD* and either *pA-uL11^{K3A}-Myc* or *pA-uL11^{K3Y}-Myc*. The FLAG-tagged chromodomain was immunoprecipitated using anti-FLAG antibodies. Contrarily to uL11-Myc that co-immunoprecipitated with FLAG-CortoCD, neither uL11K3-A-Myc nor uL11K3Y-Myc co-immunoprecipitated with it, corroborating our previous results showing that uL11K3 trimethylation mediates the interaction with CortoCD (Fig 6).

We next asked whether the uL11 mutant proteins were still able to bind chromatin. To address this question, we performed Histone Association Assays (HAA) [32]. Chromatin was extracted from S2 cells permanently expressing uL11-HA, uL11K3A-HA or uL11K3Y-HA and purified by immunoprecipitation with an anti-Histone H3 antibody. As expected, Histone H2B was co-immunoprecipitated with H3. Not only the wild type protein, but also uL11K3A and uL11K3Y mutant proteins were found associated with chromatin suggesting that their ability to modulate transcription might be kept (Fig 7).

## Transcriptomic analysis of *uL11* and *corto* mutants

We then performed a transcriptomic analysis from wing imaginal discs of third instar larvae either trans-heterozygous for two loss-of-function alleles of *corto* (*corto^{L1}/corto^{420}*) or homozygous for the *uL11* alleles (*uL11^{K3A}* or *uL11^{K3Y}*). The *w^{1118}* line was used as reference. Total numbers of reads are shown in S2 Table. Differential analyses were performed to obtain adjusted p-values associated to expression fold-changes for the three genotypes as compared to the reference. Taken a log$_2$(fold-change) < -0.50 or > 0.50 and an adjusted p-value < 0.05, we found 458 down-regulated and 481 up-regulated genes in *corto^{L1}/corto^{420}* mutants (S3 Table). Strikingly, 241 of these deregulated genes were also deregulated in wing imaginal discs overexpressing the Corto chromodomain [21] and all of them were deregulated in the same direction. Notably, 55.5% of the genes corresponding to the GO Cellular Component term "cytoplasmic translation" (66/119) were up-regulated in *corto^{L1}/corto^{420}* mutants and most of them (54/66) were also up-regulated in wing imaginal discs overexpressing the Corto chromodomain (S4 Table). These results indicated that CortoCD overexpression behaved as a dominant negative allele and confirmed that Corto down-regulates *RPGs*, directly or indirectly.

Using the same cutoffs, we found 143 down-regulated and 251 up-regulated genes in the *uL11^{K3A}* mutant. Down-regulated genes were enriched in GO terms related to transcription factor activity and sequence-specific DNA binding whereas up-regulated ones were enriched in GO terms glutathione metabolic process, telomere maintenance and DNA recombination (S4 Table). Only few genes were deregulated in *uL11^{K3Y}* (39 down-regulated and 45 up-regulated). Up-regulated genes were enriched in the category "glutathione metabolism" (KEGG pathway, Benjamini adjusted p-value 5.90E-03) as for *uL11^{K3A}*. Fifty-two deregulated genes were shared with *uL11^{K3A}* (S3 Table). Most of them were deregulated in the same direction (29 genes up-regulated and 17 genes down-regulated). Hence, contrarily to *uL11* overexpression [17], *RPGs* and *RiBi* genes were not deregulated in *uL11* mutants. Interestingly, among the 251 genes up-regulated in *uL11^{K3A}*, 82 were up-regulated in other *RPG* mutants [38, 39] (S3 Table).

Very few deregulated genes were shared between the *uL11* mutants and *corto* (69 shared by *uL11^{K3A}* and *corto*, 29 shared by *uL11^{K3Y}* and *corto*, 14 shared by the three genotypes) (Fig 8). We chose some genes to analyze their expression by RT-qPCR in the three mutants: *CG13516*,

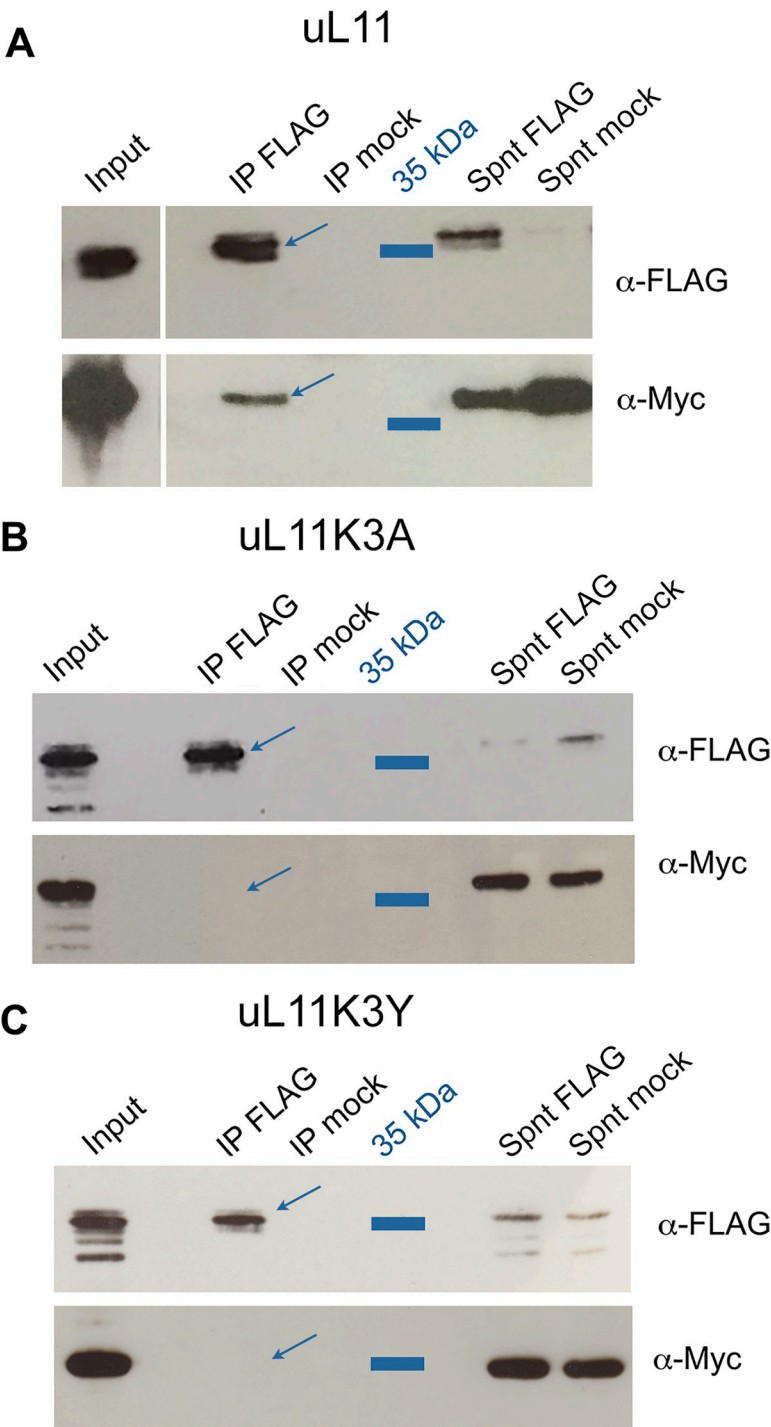

**Fig 6. uL11 but neither uL11K3A nor uL11K3Y co-immunoprecipitates with the chromodomain of Corto.** S2 cells were co-transfected with plasmids expressing FLAG-CortoCD and uL11-Myc, uL11K3A-Myc or uL11K3Y-Myc. Immunoprecipitations were performed with anti-FLAG antibodies (α-FLAG) or anti-HA antibodies (mock) and Western blot revealed using α-FLAG or anti-Myc antibodies (α-Myc). Spnt: supernatant, IP: immunoprecipitation. A - FLAG-CortoCD co-immunoprecipitated with uL11-Myc (arrow). B - FLAG-CortoCD did not co-immunoprecipitate with uL11K3A-Myc. C - FLAG-CortoCD did not co-immunoprecipitate with uL11K3Y-Myc.

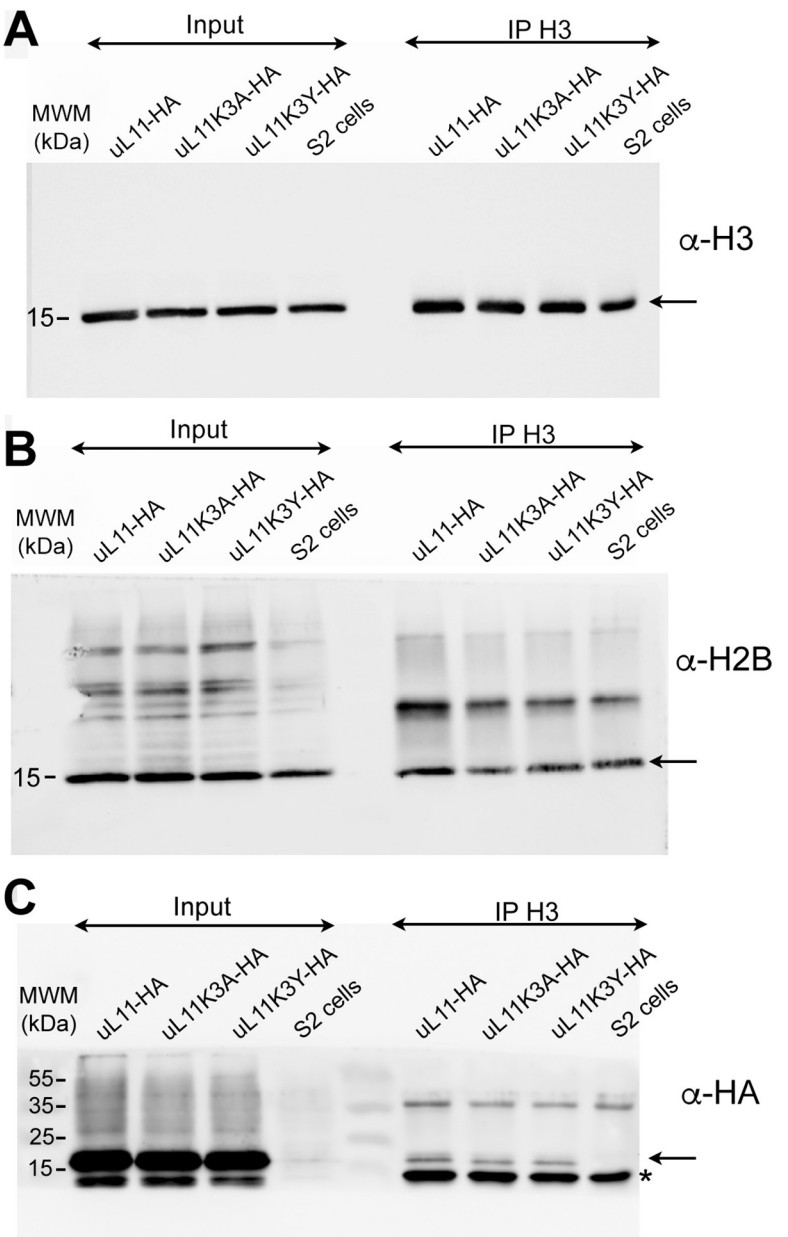

**Fig 7. uL11K3A and uL11K3Y bind chromatin.** Chromatin was extracted from S2 cells stably expressing uL11-HA, uL11K3A-HA or uL11K3Y-HA and immunoprecipitated with anti-H3 antibodies. Inputs and immunoprecipitated materials were loaded on gels for electrophoresis. Western blots were revealed using anti-H3 (A), anti-H2B (B) or anti-HA (C). The arrows show H3, H2B and the uL11-HA proteins, respectively.

one of the few genes deregulated in the three genotypes and the most down-regulated gene in $uL11^{K3Y}$, *Hsp67Bc*, encoding a small heat-shock protein involved in cold stress tolerance [40], which is up-regulated in $corto^{L1}/corto^{420}$ and $uL11^{K3Y}$, and *GstE6*, encoding a Glutathione S-transferase up-regulated in $uL11^{K3A}$ and $uL11^{K3Y}$. The RNA-seq data were confirmed by RT-qPCR for these three genes in the three mutants, except for *Hsp67Bc* for which the increase in $corto^{L1}/corto^{420}$ was not significant (Fig 8).

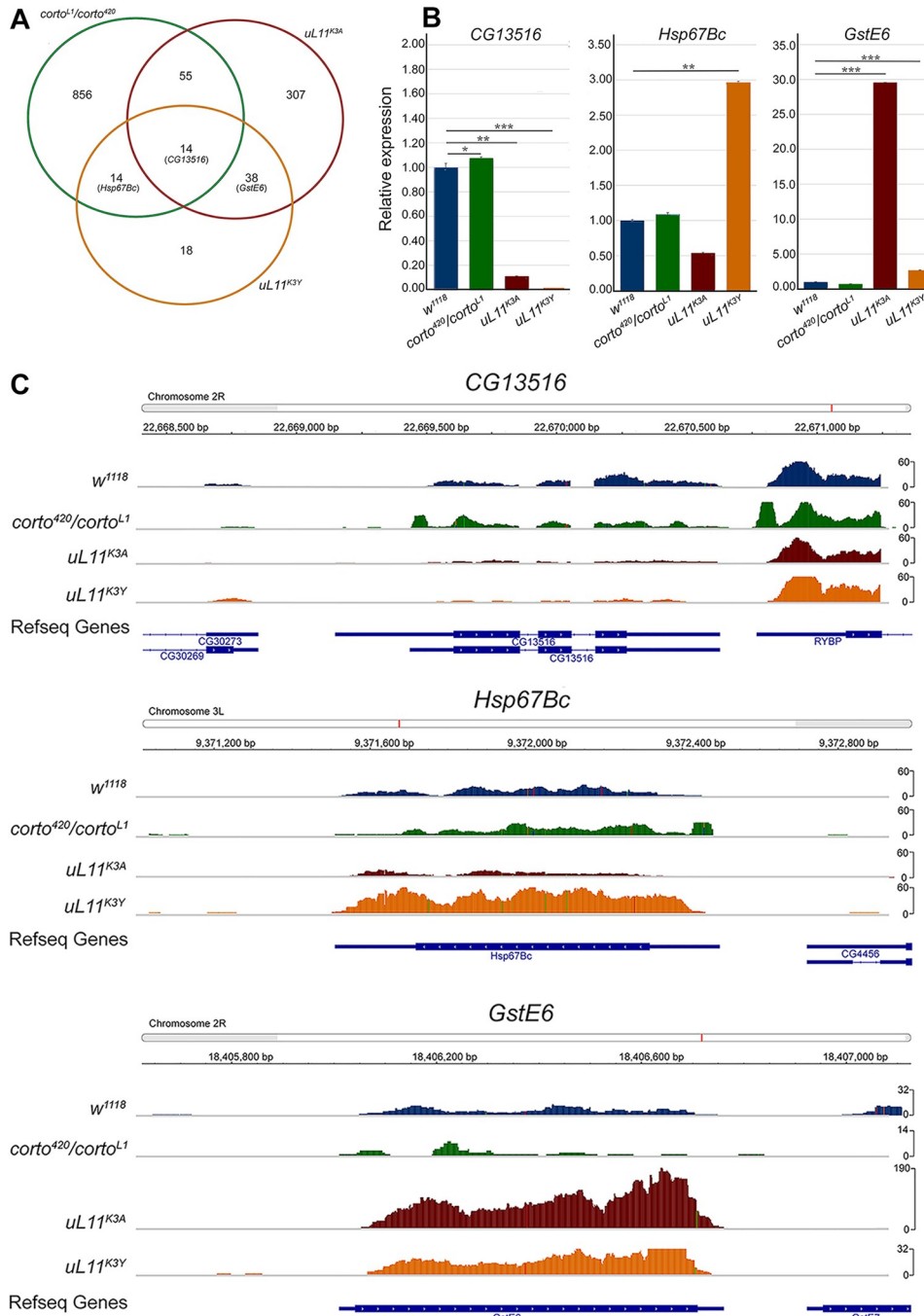

**Fig 8. RNA-seq analyses of *corto* and *uL11* mutants. A**—Venn diagrams showing the intersection of genes deregulated (up- and down-regulated) in *corto^L1^/corto^420^*, *uL11^K3A^* and *uL11^K3Y^* (cutoffs: adjusted p-value < 0.05; log2(fold-change) < -0.5 ou > 0.5). See S3 Table for detailed gene lists. The three genes validated by RT-qPCR are indicated. **B**—RT-qPCR analysis of *CG13516*, *Hsp67Bc* and *GstE6* expression in *corto^L1^/corto^420^*, *uL11^K3A^* and *uL11^K3Y^* wing imaginal discs. Expressions were normalized on the geometric mean of *GAPDH* and *Spt6*. Mean of three replicates. Error bars correspond to standard deviations. * p-value < 0.05; ** p-value < 0.01; *** p-value < 0.001. **C**– Snapshots showing the mapping of reads on *CG13516*, *Hsp67Bc* and *GstE6* in the reference line *w^1118^*, *corto^420^/corto^L1^*, *uL11^K3A^* and *uL11^K3Y^* wing imaginal discs.

## Discussion

We have previously shown that ribosomal protein uL11 interacts with the chromodomain of the *Drosophila* Enhancer of Trithorax and Polycomb Corto when tri-methylated on lysine 3 (uL11K3me3) [17]. uL11, Corto and RNA Polymerase II co-localize at many sites on polytene chromosomes and overexpression of *uL11* induces the transcription of many *RPGs* and *RiBi* genes. These data have confirmed that *Drosophila* uL11 is involved in transcription [21] and further suggest that the lysine 3 supports this extra-ribosomal activity. In the aim of testing this hypothesis, we generated mutant alleles of *uL11* using the CRISPR/Cas9 technology. By introducing a template to promote Homology-Directed Repair, we obtained a mutant in which the lysine was replaced by an alanine. However, we also obtained mutants harbouring indels probably obtained by Non-Homologous End Joining. Strikingly, the lysine 3 codon of uL11 was either deleted or substituted by another amino acid in all recovered mutants. Hence, it seems that a strong selection pressure occurs to maintain the *uL11* ORF, consistent with the haplo-insufficiency of this *RPG*.

### A single amino acid substitution generates a *Minute* phenotype

The *uL11$^{K3A}$* allele is almost totally lethal at the homozygous state. The few escapers are only males that hatch with a developmental delay larger than two days as compared to wild-type flies raised in the same conditions. Interestingly, *uL11$^{K3A}$* heterozygotes are also delayed but slightly less (about one day). In addition, *uL11$^{K3A}$* heterozygous females are frequently sterile making it necessary to carry out crosses of heterozygous *uL11$^{K3A}$* males with wild-type females and to genotype the offspring in order to maintain the stock. Hence, the *uL11$^{K3A}$* allele clearly appears dominant, which is also visible for other phenotypes, such as the shorter and thinner macrochaetes. Reduced viability, notably of females, delayed development, thin bristles and dominance are signatures of the *Minute* mutations that have been shown to correspond to *RPG* deletions [2]. The associated phenotypes are thought to reflect a defect in RPs' stoichiometry resulting in a decreased capacity for protein synthesis. In accordance, tissues that are the most dependent on translation are likely to be the most affected. For instance, high ribosome biogenesis level is suggested to be necessary for the maintenance of germinal stem cells in the *Drosophila* ovarium, which could explain the reduced fertility of *Minute* females [41, 42]. Similarly, macrochaetes are described to require a very high amount of protein synthesis over a short developmental period [2]. Unexpectedly, replacement of a single lysine by an alanine in the N-terminal tail of uL11 induces a *Minute* phenotype. The global level of translation in this mutant is decreased although the mutated protein can be efficiently incorporated into translating ribosomes. However, translation speed or accuracy might be altered. Whatever the exact origin, the *Minute* phenotype of the *uL11$^{K3A}$* mutant might be due to an alteration in translation. In yeast, *uL11* loss-of-function has been shown to halve the speed of translation and to cause increased amino acid mis-incorporation and termination codon readthrough [15]. Another possibility would be that uL11K3A ribosomes display altered affinity for specific mRNA, as has been described for ribosomes lacking RPL38/eL38 in mice [43].

Surprisingly, the *uL11$^{K3Y}$* mutant does not display any *Minute* phenotype. The N-terminal extension of uL11 is supposed to be unstructured as neither the first 6 nor the first 9 amino acids were resolved in the structure of the *D. melanogaster* and *S. cerevisiae* 80S ribosome, respectively [14, 44]. This region is composed of hydrophilic amino acids and could thus be use as a platform for protein interactions. Addition of methyl groups on lysines, even if it has no effect on the overall charge of the residue, increases its hydrophobicity. Trimethylation of uL11 lysine 3 might thus modulate the activity of the N-terminal tail [45]. However, in *S. cerevisiae* and *pombe*, deletion of Rkm2, the methyltransferase responsible for uL11 lysine 3

methylation has little impact on ribosome assembly and function or on cell viability [19, 46] suggesting that the methylation of lysine 3 is not crucial for basic translational activity. In the $uL11^{K3Y}$ mutant, the tyrosine, a hydrophobic residue, might mimic the effect of methyl groups. If this is the case, in-depth comparison of translation between $uL11^{K3Y}$ and $uL11^{K3A}$ mutants, including sequencing of polysomal mRNA, should permit to elucidate the role of the uL11 N-terminal tail methylation in translation.

## Extra-ribosomal activities of uL11

Regulation of translational capacity might be indirect and due to extra-ribosomal activities of free RPs, notably in transcriptional regulation (for a recent review see [47]). Overexpression of *uL11* increases the transcription of many *RPGs* and *RiBi* genes [21]. Although it cannot be ruled out that artificial overexpression induces a non-specific response, it is striking that over-expression of *RPGs* is also observed in loss-of-function *corto* mutants that encode a direct part-ner of uL11. Corto, by repressing *RPG* expression, could ensure that all ribosomal proteins are present at the correct stoichiometry, thus preventing ribosomal stress. As the $uL11^{K3Y}$ mutant displays neither *Minute* phenotypes nor a global decrease in translation, we assumed that solely its transcriptional activity would be affected. However, only very few genes are deregu-lated in this mutant questioning the existence of a proper transcriptional activity for uL11. To reconcile these findings, it is tempting to speculate that uL11, by physically interacting with Corto on chromatin, fine-tunes its transcriptional regulation of *RPGs*. In a context where uL11 no longer interacts with Corto, *i.e.* in the *uL11* lysine 3 mutants, the expression of these genes does not vary. Going with that model, if other ribosomal proteins were now out of stoichiome-try (for example in another *Minute* mutant), one would expect that the extra uL11 would accu-mulate, bind Corto, and repress the expression of other *RPG* genes to restore balance.

Alternatively, the absence of obvious phenotypes and the fact that only few genes are dereg-ulated in $uL11^{K3Y}$ mutant could mean that the transcriptional activity of uL11 is triggered under stress conditions which is the case for many extra-ribosomal functions of RPs. For example, in *S. cerevisiae uL11* has been shown to regulate the PHO pathway in low phosphate conditions potentially at the transcriptional level [8]. It would thus be interesting to test whether the *Drosophila uL11* mutants, and especially $uL11^{K3Y}$, display an altered resistance to stresses that affect ribosome biogenesis, for instance by raising them with specific food diets.

## Supporting information

**S1 Fig. Specificity of the anti-uL11K3me3 antibody.** 0.2 (left) and 0.05 μg (right) of each pep-tide were deposited on a nitrocellulose membrane. Membranes were then incubated with the indicated primary antibodies. Secondary antibodies were as described in Materials and Meth-ods. Peptides: unmethylated uL11, uL11K10me3, uL11K3A, uL11K3me2, and uL11K3me3 peptides were synthesized at the proteomic platform of the Institute of Biology Paris Seine; H3K4me3 and H3K9me3 peptides were from Diagenode, C16000003 and C160000056, respectively. Antibodies: PI: rabbit preimmun serum; α-uL11: 1/14000, described in Materials and Methods; α-uL11K3me3: 1/10000, described in Materials and Methods; α-H3K4me3: 1/1000, Diagenode C15310003; α-H3K9me3: 1/1000, Diagenode C15100146. Secondary anti-bodies: 1/10000.
(TIF)

**S2 Fig. Molecular screening for the $uL11^{K3A}$ allele.** A–Rationale for discriminative PCR. Pur-ple bases correspond to the target codon. Red bases stand for locked nucleic acids (LNA). The LNAWT primer ended with the lysine AAA codon of the wild-type uL11 gene whereas the

LNAK3A primer ended with the alanine GCC codon corresponding to the desired mutation. B–qPCRs were performed with the LNAK3A primer matching the uL11$^{K3A}$ allele. Red curve: plasmid carrying the uL11$^{K3A}$ allele as positive control. Black curve: genomic DNA from a wild-type fly. Blue curves: pools of up to 5 different genomic DNAs from candidate G1 flies considered to be positive. Green curves: pools of up to 5 different genomic DNAs from candidate G1 flies considered to be negative. C–The same qPCRs were performed on individual genomic DNAs from the pools that were previously found to be positive for the uL11$^{K3A}$ allele. Several individuals wearing the mutation were thus identified (blue curves). D–High Resolution Melting Analysis (HMRA) of *uL11* mutants. Melting profile of the *uL11* amplicons from genomic DNAs of G1 flies. Melting peaks flatter and broader than the reference (black) revealed the presence of two different amplicons, indicating that the tested DNA contained a mutation at the *uL11* locus. Melting curves were normalized according to the method described by [27]. RFU: Relative Fluorescence Unit.
(TIF)

**S3 Fig. Sequence of the *uL11* alleles of G0 flies.** Founder G0 flies were named after their gender (M, male; F, female) and the order of their emergence. Each allele was recovered in several descendants of the same founders. The *uL11$^{K3A}$* and *uL11$^{ΔK3}$* alleles were found in the progeny of three and two different founders, respectively. Substitution alleles were named to reflect the amino acid change in the uL11 protein, following the amino acid one letter code. The bottom two alleles were named after the reading frameshift they introduce in the *uL11* gene. The wild-type *uL11* sequence is provided as reference. The start and the lysine 3 codons of *uL11* are highlighted in grey. Mutants F-4 and F+2 introduce a +2 reading frame shift that puts the uL11 CDS in frame with an ATG codon located in the 5'UTR. A protein with a 24 amino acid extension might then be produced.
(TIF)

**S4 Fig. Analysis of the *uL11$^{K3A-12}$* mutant.** A—From left to right: thorax of a wild-type female, a *uL11$^{K3A-12}$*/*uL11$^{+}$* heterozygous female, and a *uL11$^{K3A-12}$*/*uL11$^{K3A-12}$* homozygous female. Anterior and posterior scutellar bristles are colorized. B–Developmental time of the *uL11$^{K3A-12}$* mutant as compared to *uL11$^{K3A-43}$*—the mutant presented in the main text. The percentage of flies emerged from day 8 to 15 is shown. The total number of flies emerged is indicated in the legend. Solid lines: heterozygous flies; dotted line: homozygous flies; blue: *uL11$^{+}$*, burgundy: *uL11$^{K3-43}$*, pink: *uL11$^{K3A-12}$*.
(TIF)

**S5 Fig. Analysis of bristles and wings in female *uL11* mutants.** A–Length of anterior scutellar bristles of wild-type females (blue; n = 54), *uL11$^{K3A}$*/*uL11$^{+}$* (burgundy, n = 33) and *uL11$^{K3Y}$*/*uL11$^{K3Y}$* (orange, n = 25). B–Length of posterior scutellar bristles of wild-type males (blue; n = 51), *uL11$^{K3A}$*/*uL11$^{+}$* (burgundy, n = 36) and *uL11$^{K3Y}$*/*uL11$^{K3Y}$* (orange, n = 50). C–Wing size of uL11 wild-type females (blue; n = 29), *uL11$^{K3A}$*/*uL11$^{+}$* (burgundy, n = 30) and *uL11$^{K3Y}$*/*uL11$^{K3Y}$* (orange, n = 25). t-tests: *** $p$-value $< 0.001$; ** $p$-value $< 0.01$; * $p$-value $< 0.05$; ns: non significant.
(TIF)

**S1 Table. Oligonucleotides used in this study.** In primers LNA-WT and LNA-K3A, uppercase nucleotides correspond to the LNA bases. In primers pho-sgRNA_F and pho-sgRNA_R, uppercase nucleotides correspond to the floating sequences used for cloning. The bold guanosine was introduced to increase efficiency of the U6 promoter. In the ssODN, the complementary ATG and alanine codon sequences are bold and in uppercases, the PAM sequence

corresponding to the single guide RNA is in bold.
(PDF)

**S2 Table. RNA-seq of wing imaginal discs (GEO accession number GSE181926).**
(PDF)

**S3 Table. Genes deregulated in at least one of the three genotypes (*corto^L1^ /corto^420^*, *uL11^K3A^*, *uL11^K3Y^*).** Green: Up-regulated genes, log2 fold-change > 0.5, adjusted p-value < 5. E-02. Orange: Down-regulated genes, log2 fold-change < -0.5, adjusted p-value < 5.E-02. Blue: Genes up-regulated in other RPG mutants [35, 36].
(PDF)

**S4 Table. Ontology of genes deregulated in *corto^L1^/corto^420^* and *uL11^K3A^* mutants.** CC: Cellular Component; BP: Biological Process; MF: Molecular Function.
(PDF)

**S1 Raw images.**
(PDF)

# Acknowledgments

We thank the members of the team for stimulating discussions, Jean-Michel Gibert for critical reading of the manuscript, Immane R'Kiki for technical assistance, Naïra Naouar from the ARTbio Bioinformatics platform (IBPS) for the training of HG in NGS analyses, the Bloomington Stock Center for fly strains.

# Author Contributions

**Conceptualization:** Jérôme Deraze, Sébastien Bloyer, Emmanuèle Mouchel-Vielh, Frédérique Peronnet, Hélène Thomassin.

**Data curation:** Frédérique Peronnet.

**Formal analysis:** Frédérique Peronnet, Hélène Thomassin.

**Funding acquisition:** Frédérique Peronnet.

**Investigation:** Héloïse Grunchec, Jérôme Deraze, Delphine Dardalhon-Cuménal, Valérie Ribeiro, Anne Coléno-Costes, Karine Dias, Frédérique Peronnet, Hélène Thomassin.

**Methodology:** Héloïse Grunchec, Jérôme Deraze, Delphine Dardalhon-Cuménal, Valérie Ribeiro, Anne Coléno-Costes, Karine Dias, Sébastien Bloyer, Frédérique Peronnet, Hélène Thomassin.

**Project administration:** Frédérique Peronnet.

**Resources:** Frédérique Peronnet.

**Software:** Jérôme Deraze.

**Supervision:** Sébastien Bloyer, Frédérique Peronnet, Hélène Thomassin.

**Validation:** Frédérique Peronnet, Hélène Thomassin.

**Visualization:** Héloïse Grunchec, Jérôme Deraze, Emmanuèle Mouchel-Vielh, Frédérique Peronnet.

**Writing – original draft:** Héloïse Grunchec, Jérôme Deraze, Sébastien Bloyer, Emmanuèle Mouchel-Vielh, Frédérique Peronnet, Hélène Thomassin.

**Writing – review & editing:** Emmanuèle Mouchel-Vielh, Frédérique Peronnet, Hélène Thomassin.

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
