## [Decision Letter · Decision Letter 0]

22 Apr 2022

PONE-D-22-02062Single amino-acid mutation in a Drosophila melanogaster ribosomal protein: an insight in uL11 transcriptional activityPLOS ONE

Dear Dr. Peronnet,

Thank you for submitting your manuscript to PLOS ONE. After careful consideration, we feel that it has merit but does not fully meet PLOS ONE’s publication criteria as it currently stands. Therefore, we invite you to submit a revised version of the manuscript that addresses the points raised during the review process.

 First, I apologise for the extremely long time it has taken to review this paper. I had problems finding willing reviewers. However, I now have one detailed reviewer's report and I have read the paper carefully myself. I generally agree with the comments this reviewer has made and would ask you to address them. However, in point (1) they state  "I would like however to see a second independent K3A line assessed on the major phenotypes (for instance developmental delay and Minute bristles) to rule out genetic background issues (especially since K3Y, or the other double substitutions do not give the same phenotype)". I agree that ruling out genetic background issues is important, however, if no second independent K3A line is available, this could be done by recombining the K3A mutation in to alternative genetic backgrounds using using flanking genetic markers.  Some other minor points:The sentence (final paragraph of the introduction) "We describe here the uL11K3A and uL11K3Y mutants in which the lysine 3 codon of uL11 is replaced by a non-methylable alanine and a tyrosine codon, respectively." is very difficult to follow and should be re-written along the lines of: "In this study we have replaced the lysine 3 codon of uL11 with codons for amino acids that are not subject to methylation; alanine (uL11K3A) and tyrosine (uL11K3Y)."Results, line 2, first paragraph:"uL11 is located within a cluster of highly transcribed genes, many of which are also essential (eIF5A, RpL39/eL39, yki...)" - please rewrite as "uL11 is located within a cluster of highly transcribed genes, many of which are also essential (including eIF5A, RpL39/eL39 and yki)." 

Please include a reference describing the "single step CRISPR/Cas9 mediated HDR using a single-stranded oligodeoxynucleotide donor template (ssODN)" that you based your experimental strategy on at the end of the first paragraph of the Results. 

With respect to the publication criteria, I am happy to confirm that in my opinion, the paper presents the results of original research that has not been reported elsewhere, the experiments, statistics, and other analyses are generally performed to a high technical standard and are described in sufficient detail, the article is presented in an intelligible fashion and is written in standard English, the research meets all applicable standards for the ethics of experimentation and research integrity and that article adheres to appropriate reporting guidelines and community standards for data availability. Generally the conclusions are appropriate, however, there is some scope to make the conclusions from the RNA-seq studies clearer and to propose alternative models to explain the data. 

We look forward to receiving your revised manuscript.

Kind regards,

Barbara Jennings

Academic Editor

PLOS ONE

Journal Requirements:

“This work was funded by the Centre National de la Recherche Scientifique (CNRS), Sorbonne University, and by a Fondation ARC grant to FP (PJA20171206407). HG was funded by a doctoral fellowship from the MESRI (Ministère de l’Enseignement Supérieur, de la Recherche et de l’Innovation) and a 4th year doctoral fellowship from Fondation ARC (ARCDOC42020020001381). JD was funded by a doctoral fellowship from the MESRI and a 4th year doctoral fellowship from the Fondation pour la Recherche médicale (FDT20160435164). This work was supported by the France Génomique national infrastructure, funded as part of the "Investissements d'Avenir" program managed by the Agence Nationale de la Recherche (contract ANR-10-INBS-0009).”

“FP: Centre National de la Recherche Scientifique (CNRS), Sorbonne University, Fondation ARC grant (PJA20171206407)

HG: Doctoral fellowship from the MESRI and Fondation ARC (ARCDOC42020020001381)

JD: Doctoral fellowship from the MESRI and Fondation pour la Recherche médicale (FDT20160435164)

In your cover letter, please note whether your blot/gel image data are in Supporting Information or posted at a public data repository, provide the repository URL if relevant, and provide specific details as to which raw blot/gel images, if any, are not available. Email us at plosone@plos.org if you have any questions

6. We noticed you have some minor occurrence of overlapping text with the following previous publication(s), which needs to be addressed:

- https://tel.archives-ouvertes.fr/tel-01878354/file/2017PA066342.pdf

In your revision ensure you cite all your sources (including your own works), and quote or rephrase any duplicated text outside the methods section. Further consideration is dependent on these concerns being addressed

Reviewers' comments:

Reviewer's Responses to Questions

**Comments to the Author**

1. Is the manuscript technically sound, and do the data support the conclusions?

Reviewer #1: Yes

2. Has the statistical analysis been performed appropriately and rigorously? 

Reviewer #1: Yes

3. Have the authors made all data underlying the findings in their manuscript fully available?

Reviewer #1: Yes

4. Is the manuscript presented in an intelligible fashion and written in standard English?

Reviewer #1: Yes

5. Review Comments to the Author

Reviewer #1: Review PlosOne Manuscript PONE-D-22-02062

In this manuscript, Grunchec et al. report for the first time the generation and characterization of two new mutant lines for the ribosomal protein uL11/ Rpl12. This factor located at the basis of the ribosome P-Stalk plays an essential role in translation efficiency. A previous work from the same group demonstrated that uL11 displays extra-ribosomal activities and is able to bind chromatin and enhance the transcription of others Ribosomal Protein Genes (RPGs) and Ribosomal Biogenesis genes (RiBis) when over-expressed. Moreover, they found that uL11 is trimethylated on lysine 3 allowing its interaction with the chromodomain of the Enhancer of Polycomb and Trithorax Corto. These findings let the authors postulate that the transcriptional activity of uL11 is controlled by this Lysine 3 methylation. To test this hypothesis, they generated in the present study, point mutations of this amino acid using a CRISPR/Cas9 strategy. They obtained several mutant lines and focused their work on 2 mutations, namely K3A (lysine to alanine) and K3Y (lysine to tyrosine).

The first part of the manuscript describes in great details the making and the validation of the mutants. The experiments are well designed and perfectly controlled, and show clearly that the lines obtained carry the right mutations.

The second part focuses on the phenotypic characterization of these two mutants. Both are found to have lost their ability to interact with Corto. Interestingly, the K3A lines behaves as a typical Minute mutant. Surprisingly, K3Y does not display such phenotype.

In order to explain this discrepancy, they next ask whether these mutations impact on the translational and transcriptional activities of uL11. They show that both are well associated with the ribosome. However, while K3A mutant display an alteration of the yield of translation, K3Y behaves as Wild Type.

Finally, the author performed a transcriptomic analysis from third instar wing discs of Corto loss of function mutants and both uL11 Lysine 3 mutants. Cross comparison of RNA expression profiles show that very few deregulated genes were shared between the 3 conditions.

Overall, the experiments performed in this manuscript are thoughtful and well implemented. Before I can fully support publication, there are just a few key experiments I would like to see to back the claims of the authors, or a few points that need to be rephrased or discussed.

1) The results obtained in the mutant validation and the phenotypic description part of the manuscript are convincing and very detailed. I would like however to see a second independent K3A line assessed on the major phenotypes (for instance developmental delay and Minute bristles) to rule out genetic background issues (especially since K3Y, or the other double substitutions do not give the same phenotype).

2) The K3A mutants display a clear translational defect (Minute phenotype and lower translation rate). The K3A is clearly dominant in all assays performed by the authors, and it is clearly stated as such in the text. However, how can the authors explain that they did not see any change in translation in the heterozygous K3A/uL11 but only in the homozygous K3A, even though the heterozygous already has clear “Minute-like” phenotypes (Fig3&4 vs Fig 5 and text page 12). Either they need to perform a better quantification, or they need to discuss how this could be. I suggest that the authors compare with at least one other known Minute mutant which would help to understand what could be expected and what is the level of sensitivity of the tests they have used to measure translational efficiency.

3) The conclusions regarding the potential role of uL11 as a transcriptional regulator are rather confusing when compared to the findings published by the same group.

Despite the fact that the Lysine 3 mutants do not interact anymore with Corto, almost none of the RPG and RiBis genes found to be deregulated previously (by uL11 over-expression) seem to be affected by these mutations. The authors hypothesize that the pool of free mutated uL11 available for extra- ribosomal functions in transcriptional regulation could be reduced. While technically difficult to verify such hypothesis, the authors should check whether these mutants are still bound to the chromatin and to which extent compared to WT uL11 (uL11 antibodies are available).

4) K3Y is an interesting mutant as it does not seem to lose its translational activity and could constitute a good candidate to decipher the transcriptional role of uL11. However, I feel the transcriptomic analyses shown here weaken the paper rather than strengthen it. It adds observations without clear direction appear preliminary and are presented in a slightly confusing way. My suggestion would be to refocus that part on the clear result here: uL11 K3 mutants have hardly any transcriptional effects, and are different from corto mutations. Unlike what was observed for the over-expressed proteins, no clear changes were observed, in particular to Ribosomal protein genes.

Regarding the exploitation of the transcriptomic data I have a few comments that highlight the impression the data is preliminary.

In Supplemental Table 2: I do not understand the numbers presented and how the number of aligned reads can be bigger than the number of total reads for some samples? Similarly how can the number of unmapped reads be bigger than the number of total reads?

In Supplemental Table 3, I do not understand what the blue highlight is for?

In Supplemental Table 5, what would be the model compatible with the observation that the Mad sites are found in the vicinity of both up and down regulated genes in roughly equal proportions? Is that really biologically informative?

I feel these extra analyses on the transcriptome are too preliminary, and would be better left out.

5) Finally, the discussion on the absence of transcriptional change in uL11K3A or uL11K3Y should be more direct and clearly discuss the models for the discrepancy between gain and loss of function, for instance by being open to the caveats of gain of function experiments. One interesting model that is not discussed, or not spelled out clearly enough, is that Corto could be a regulator of Ribosomal protein gene expression, ensuring that all proteins are present at correct stoichiometry. Contrary to the gain of function of uL11, in the case of uL11K3A or uL11K3Y mutations, the stoichiometry is not affected, hence no transcriptional changes. However, going with that model, if other Ribosomal protein were now out of stoichiometry (Minute mutation), one would expect that the extra uL11 would accumulate, bind Corto, and repress the expression of other Ribosomal protein genes to restore balance. If uL11 is critical here for this Corto function, other Minute mutations would not lead to Ribosomal protein gene repression in the context of the uL11 K3A or uL11 K3Y mutants as they would be incapable of binding and enhancing Corto repressive action. This model could reconcile the gain and loss of function observations and should be testable with a recombinant between uL11K3Y (fully viable mutant) and another “Minute” gene mutant on the 2nd chromosome. This model or any other that could explain the discrepancies with the previous published work of the same team should be discussed.

6. PLOS authors have the option to publish the peer review history of their article (what does this mean?). If published, this will include your full peer review and any attached files.

Reviewer #1: No

---

## [Author Response · Author response to Decision Letter 0]

27 Jul 2022

RESPONSE TO THE EDITOR

1 - I generally agree with the comments this reviewer has made and would ask you to address them. However, in point (1) they state "I would like however to see a second independent K3A line assessed on the major phenotypes (for instance developmental delay and Minute bristles) to rule out genetic background issues (especially since K3Y, or the other double substitutions do not give the same phenotype)". I agree that ruling out genetic background issues is important, however, if no second independent K3A line is available, this could be done by recombining the K3A mutation in to alternative genetic backgrounds using using flanking genetic markers. 

Answer: Two other independent K3A lines, uL11K3A-6 and uL11K3A-12 were obtained (see Supplementary Figure 3) that exhibit the same phenotypes than the uL11K3A-43 line presented in the main text. To illustrate this point, we added pictures showing Minute bristles and the analysis of developmental time of the uL11K3A-12 mutant (new Figure: Supplementary Figure 4).

2 - The sentence (final paragraph of the introduction) "We describe here the uL11K3A and uL11K3Y mutants in which the lysine 3 codon of uL11 is replaced by a non-methylable alanine and a tyrosine codon, respectively." is very difficult to follow and should be re-written along the lines of: "In this study we have replaced the lysine 3 codon of uL11 with codons for amino acids that are not subject to methylation; alanine (uL11K3A) and tyrosine (uL11K3Y)."

Answer: The sentence has been replaced (see lines 113 to 115 on the manuscript with visible corrections).

3 - Results, line 2, first paragraph:

"uL11 is located within a cluster of highly transcribed genes, many of which are also essential (eIF5A, RpL39/eL39, yki...)" - please rewrite as "uL11 is located within a cluster of highly transcribed genes, many of which are also essential (including eIF5A, RpL39/eL39 and yki)." 

Answer: The sentence has been replaced (see lines 342 and 343 on the manuscript with visible corrections).

4 - Please include a reference describing the "single step CRISPR/Cas9 mediated HDR using a single-stranded oligodeoxynucleotide donor template (ssODN)" that you based your experimental strategy on at the end of the first paragraph of the Results. 

Answer: The following reference has been added (reference 23, line 350 on the manuscript with visible corrections):

Gratz, Scott J., Alexander M. Cummings, Jennifer N. Nguyen, Danielle C. Hamm, Laura K. Donohue, Melissa M. Harrison, Jill Wildonger, and Kate M. O’Connor-Giles. (2013) Genome Engineering of Drosophila with the CRISPR RNA-Guided Cas9 Nuclease.’ Genetics 194: 1029–35. https://doi.org/10.1534/genetics.113.152710.

5 - With respect to the publication criteria, I am happy to confirm that in my opinion, the paper presents the results of original research that has not been reported elsewhere, the experiments, statistics, and other analyses are generally performed to a high technical standard and are described in sufficient detail, the article is presented in an intelligible fashion and is written in standard English, the research meets all applicable standards for the ethics of experimentation and research integrity and that article adheres to appropriate reporting guidelines and community standards for data availability. Generally the conclusions are appropriate, however, there is some scope to make the conclusions from the RNA-seq studies clearer and to propose alternative models to explain the data. 

Answer: We rewrote the discussion about RNA-seq as suggested by the reviewer (see lines 690 to 706 on the manuscript with visible corrections).

JOURNAL REQUIREMENTS

Answer: This has been done

“FP: Centre National de la Recherche Scientifique (CNRS), Sorbonne University, Fondation ARC grant (PJA20171206407)

HG: Doctoral fellowship from the MESRI and Fondation ARC (ARCDOC42020020001381)

JD: Doctoral fellowship from the MESRI and Fondation pour la Recherche médicale (FDT20160435164)

Answer: We agree with this funding statement.

We did not observe any lethality during the pupal life for all genotypes (data not shown).

Answer: The phrase has been removed (line 406 on the manuscript with visible corrections).

Answer: This has been done.

5. In your cover letter, please note whether your blot/gel image data are in Supporting Information or posted at a public data repository, provide the repository URL if relevant, and provide specific details as to which raw blot/gel images, if any, are not available. Email us at plosone@plos.org if you have any questions

Answer: The file S1_raw_images.pdf contains all the raw images.

6. We noticed you have some minor occurrence of overlapping text with the following previous publication(s), which needs to be addressed:

- https://tel.archives-ouvertes.fr/tel-01878354/file/2017PA066342.pdf

In your revision ensure you cite all your sources (including your own works), and quote or rephrase any duplicated text outside the methods section. Further consideration is dependent on these concerns being addressed.

Answer: This is the PhD thesis manuscript of the co-first author Jérôme Deraze. The reference has been added in the text (see reference 34). 

Answer: This has been done.

RESPONSE TO REVIEWER #1

In this manuscript, Grunchec et al. report for the first time the generation and characterization of two new mutant lines for the ribosomal protein uL11/ Rpl12. This factor located at the basis of the ribosome P-Stalk plays an essential role in translation efficiency. A previous work from the same group demonstrated that uL11 displays extra-ribosomal activities and is able to bind chromatin and enhance the transcription of others Ribosomal Protein Genes (RPGs) and Ribosomal Biogenesis genes (RiBis) when over-expressed. Moreover, they found that uL11 is trimethylated on lysine 3 allowing its interaction with the chromodomain of the Enhancer of Polycomb and Trithorax Corto. These findings let the authors postulate that the transcriptional activity of uL11 is controlled by this Lysine 3 methylation. To test this hypothesis, they generated in the present study, point mutations of this amino acid using a CRISPR/Cas9 strategy. They obtained several mutant lines and focused their work on 2 mutations, namely K3A (lysine to alanine) and K3Y (lysine to tyrosine).

The first part of the manuscript describes in great details the making and the validation of the mutants. The experiments are well designed and perfectly controlled, and show clearly that the lines obtained carry the right mutations.

The second part focuses on the phenotypic characterization of these two mutants. Both are found to have lost their ability to interact with Corto. Interestingly, the K3A lines behaves as a typical Minute mutant. Surprisingly, K3Y does not display such phenotype.

In order to explain this discrepancy, they next ask whether these mutations impact on the translational and transcriptional activities of uL11. They show that both are well associated with the ribosome. However, while K3A mutant display an alteration of the yield of translation, K3Y behaves as Wild Type.

Finally, the author performed a transcriptomic analysis from third instar wing discs of Corto loss of function mutants and both uL11 Lysine 3 mutants. Cross comparison of RNA expression profiles show that very few deregulated genes were shared between the 3 conditions.

Overall, the experiments performed in this manuscript are thoughtful and well implemented. Before I can fully support publication, there are just a few key experiments I would like to see to back the claims of the authors, or a few points that need to be rephrased or discussed.

1) The results obtained in the mutant validation and the phenotypic description part of the manuscript are convincing and very detailed. I would like however to see a second independent K3A line assessed on the major phenotypes (for instance developmental delay and Minute bristles) to rule out genetic background issues (especially since K3Y, or the other double substitutions do not give the same phenotype).

Answer: Two others independent K3A line, uL11K3A-6 and uL11K3A-12 had been obtained (see Supplementary Figure 3). We added pictures showing Minute bristles and the analysis of developmental time of uL11K3A-12 that shows the same phenotype than the uL11K3A-43 mutant presented in the main text (see new Figure: Supplementary Figure 4).

2) The K3A mutants display a clear translational defect (Minute phenotype and lower translation rate). The K3A is clearly dominant in all assays performed by the authors, and it is clearly stated as such in the text. However, how can the authors explain that they did not see any change in translation in the heterozygous K3A/uL11 but only in the homozygous K3A, even though the heterozygous already has clear “Minute-like” phenotypes (Fig3&4 vs Fig 5 and text page 12). Either they need to perform a better quantification, or they need to discuss how this could be. I suggest that the authors compare with at least one other known Minute mutant which would help to understand what could be expected and what is the level of sensitivity of the tests they have used to measure translational efficiency.

Answer: Indeed, we did not observe any change in global translation of heterozygous uL11K3A/+ larvae although we did observe Minute phenotypes (developmental delay, shorter bristles etc.). On the one hand, the phenotypes of uL11K3A/ uL11+ flies are not as severe as those of uL11K3A/ uL11K3A, and a low decrease in global translation might be undetectable in our assays. On the other hand, we measured the translational rate in total larvae and it is known that the amount of protein synthesis varies depending on the tissue. Hence, some tissues might be more sensitive to RPG reduction than others. For example, in mouse Rpl24Bst/+ haematopoietic cells, measure of protein synthesis based on in vivo OP-Puro incorporation shows that it is highly dependent on the cell type, being decreased in granulocyte-macrophage progenitors whereas unchanged in most other progenitors (Signer, 2004). In Drosophila, many studies have been done by in vivo incorporation of OP-puro in wing imaginal discs of third instar larvae in a context of cell competition - clones of Rp-/+ cells in a Rp+/+ genetic background (Ji, 2019; Kiparaki, 2021). These are proliferating cells that might need more protein synthesis than polyploid tissues that are in the majority in larvae. These argues were added in the manuscript (see lines 475 to 484 in the manuscript with visible corrections).

3) The conclusions regarding the potential role of uL11 as a transcriptional regulator are rather confusing when compared to the findings published by the same group.

Despite the fact that the Lysine 3 mutants do not interact anymore with Corto, almost none of the RPG and RiBis genes found to be deregulated previously (by uL11 over-expression) seem to be affected by these mutations. The authors hypothesize that the pool of free mutated uL11 available for extra- ribosomal functions in transcriptional regulation could be reduced. While technically difficult to verify such hypothesis, the authors should check whether these mutants are still bound to the chromatin and to which extent compared to WT uL11 (uL11 antibodies are available).

Answer: We thank the reviewer for this comment and we agree that it was crucial to test the binding of the mutant proteins to chromatin. Unfortunately, our uL11 antibodies were not immunoprecipitating. We thus performed Histone Association Assays as described by Ricke and Bielinsky [Biol. Proced. Online 2005; 7(1): 60-69. doi:10.1251/bpo106]. We extracted chromatin from our permanently S2 cell lines expressing either uL11-HA, uL11K3A-HA or uL11K3Y-HA and purified it by immunoprecipitation with an anti-Histone 3 antibody. We checked that Histone H2B co-immunoprecipitated with H3, showing that we successfully prepared chromatin. We found that all three uL11 proteins, i.e. not only uL11 wild-type, but also uL11K3A and uL1K3Y, were present on chromatin with no detectable difference between them in the amount of associated protein at this scale. This suggests that the mutated proteins might still have the ability to regulate transcription. We added these new results in the manuscript (see text lines 538 to 544 in the manuscript with visible corrections and the new figure 7).

4) K3Y is an interesting mutant as it does not seem to lose its translational activity and could constitute a good candidate to decipher the transcriptional role of uL11. However, I feel the transcriptomic analyses shown here weaken the paper rather than strengthen it. It adds observations without clear direction appear preliminary and are presented in a slightly confusing way. My suggestion would be to refocus that part on the clear result here: uL11 K3 mutants have hardly any transcriptional effects, and are different from corto mutations. Unlike what was observed for the over-expressed proteins, no clear changes were observed, in particular to Ribosomal protein genes.

Regarding the exploitation of the transcriptomic data I have a few comments that highlight the impression the data is preliminary.

In Supplemental Table 2: I do not understand the numbers presented and how the number of aligned reads can be bigger than the number of total reads for some samples? Similarly how can the number of unmapped reads be bigger than the number of total reads? 

Answer: We thank the reviewer for this remark and we apologize for this mistake. A table with the right numbers is now presented (see Sup Table 2).

In Supplemental Table 3, I do not understand what the blue highlight is for?

Answer: As indicated in the legend of Sup Table 3, these genes are those which are up-regulated in other RPG mutants (described in references 38 and 39).

In Supplemental Table 5, what would be the model compatible with the observation that the Mad sites are found in the vicinity of both up and down regulated genes in roughly equal proportions? Is that really biologically informative?

I feel these extra analyses on the transcriptome are too preliminary, and would be better left out.

Answer: We understand the reviewer's point of view and we have eliminated this part. (see linesd 603 to 614 and 707 to 717 of the manuscript with visible corrections).

5) Finally, the discussion on the absence of transcriptional change in uL11K3A or uL11K3Y should be more direct and clearly discuss the models for the discrepancy between gain and loss of function, for instance by being open to the caveats of gain of function experiments.

 One interesting model that is not discussed, or not spelled out clearly enough, is that Corto could be a regulator of Ribosomal protein gene expression, ensuring that all proteins are present at correct stoichiometry. Contrary to the gain of function of uL11, in the case of uL11K3A or uL11K3Y mutations, the stoichiometry is not affected, hence no transcriptional changes. However, going with that model, if other Ribosomal protein were now out of stoichiometry (Minute mutation), one would expect that the extra uL11 would accumulate, bind Corto, and repress the expression of other Ribosomal protein genes to restore balance.

If uL11 is critical here for this Corto function, other Minute mutations would not lead to Ribosomal protein gene repression in the context of the uL11 K3A or uL11 K3Y mutants as they would be incapable of binding and enhancing Corto repressive action.

This model could reconcile the gain and loss of function observations and should be testable with a recombinant between uL11K3Y (fully viable mutant) and another “Minute” gene mutant on the 2nd chromosome. This model or any other that could explain the discrepancies with the previous published work of the same team should be discussed.

Answer: We thank the reviewer for these remarks. We have modified the discussion to take them into account. In particular, we have focused on the fact that few genes are deregulated in the uL11K3Y mutant and that the interaction between Corto and uL11 could allow to fine-tune RPGs in order to ensure the stoichiometry of ribosomal proteins (see lines 690 to 706 of the manuscript with visible corrections).

---

## [Editor Report · Decision Letter 1]

4 Aug 2022

Single amino-acid mutation in a Drosophila melanogaster ribosomal protein: an insight in uL11 transcriptional activity

PONE-D-22-02062R1

Dear Dr. Peronnet,

We’re pleased to inform you that your manuscript has been judged scientifically suitable for publication and will be formally accepted for publication once it meets all outstanding technical requirements.

Kind regards,

Barbara Jennings

Academic Editor

PLOS ONE
---

## [Editor Report · Acceptance letter]

9 Aug 2022

PONE-D-22-02062R1 

Single amino-acid mutation in a Drosophila melanogaster ribosomal protein: an insight in uL11 transcriptional activity 

Dear Dr. Peronnet:

I'm pleased to inform you that your manuscript has been deemed suitable for publication in PLOS ONE. Congratulations! Your manuscript is now with our production department. 

Kind regards, 

on behalf of

Dr. Barbara Jennings 

Academic Editor

PLOS ONE